# Omicron infection following vaccination enhances a broad spectrum of immune responses dependent on infection history

Pronounced immune escape by the SARS-CoV-2 Omicron variant has resulted in many individuals possessing hybrid immunity, generated through a combination of vaccination and infection. Concerns have been raised that omicron breakthrough infections in triple-vaccinated individuals result in poor induction of omicron-specific immunity, and that prior SARS-CoV-2 infection is associated with immune dampening. Taking a broad and comprehensive approach, we characterize mucosal and blood immunity to spike and non-spike antigens following BA.1/BA.2 infections in triple mRNA-vaccinated individuals, with and without prior SARS-CoV-2 infection. We find that most individuals increase BA.1/BA.2/BA.5-specific neutralizing antibodies following infection, but confirm that the magnitude of increase and post-omicron titres are higher in the infection-naive. In contrast, significant increases in nasal responses, including neutralizing activity against BA.5 spike, are seen regardless of infection history. Spike-specific T cells increase only in infection-naive vaccinees; however, post-omicron T cell responses are significantly higher in the previously-infected, who display a maximally induced response with a highly cytotoxic CD8+ phenotype following their 3rd mRNA vaccine dose. Responses to non-spike antigens increase significantly regardless of prior infection status. These findings suggest that hybrid immunity induced by omicron breakthrough infections is characterized by significant immune enhancement that can help protect against future omicron variants.

Since its initial description in November 2021, the B.1.1.529 (omicron) variant of severe acute respiratory syndrome coronavirus 2 (SARS-CoV-2) has rapidly spread throughout the world[1,2]. Several omicron-lineage viruses have since emerged, with waves dominated initially by BA.1 and BA.2 variants, followed by BA.4/5, and more recently by combinations of omicron variants such as BA.2.75, BQ.1, and XBB[3]. The unprecedented number of spike mutations in omicron viruses has led to considerable immune escape from vaccine- and infection-induced immunity[4,5]. Vaccine effectiveness against symptomatic SARS-CoV-2 infection with B.1.1.529 after a third BNT162b2 mRNA vaccine dose is estimated to be 67.2% at 2–4 weeks, falling to 45.7% after 10 weeks[6]. As a result, a large number of individuals in highly vaccinated populations now have so-called hybrid immunity, generated through a combination of vaccination and infection. A multi-region cohort study in Switzerland estimated that at least 51% of the population had hybrid immunity by July 2022, with 85% having some degree of omicron-specific antibody immunity[7].

Until the widespread circulation of omicron, individuals with hybrid immunity were primarily those who were infected during SARS-CoV-2 B.1.1.7/alpha or pre-alpha ('ancestral') waves, prior to commencing their vaccine courses. These 'previously-infected' individuals have higher spike-specific serum antibody and T-cell responses after each vaccine dose compared to infection-naive vaccinees[8–10]. Hybrid immunity generated by post-vaccination infections may be

✉ e-mail: paul.klenerman@ndm.ox.ac.uk; t.desilva@sheffield.ac.uk

quantitatively and qualitatively different from responses seen in individuals who experienced SARS-CoV-2 infection before receiving a vaccination course. This may be due to differences in the priming SARS-CoV-2 exposure or lower antigenic exposure during the attenuated disease course of omicron viruses; although it is difficult to tease apart the contributions of viral phenotype change from those of pre-existing immunity[11]. Recent reports have suggested that omicron-specific immunity generated by breakthrough infections may be muted in triple-vaccinated individuals, with those with a history of prior SARS-CoV-2 having a particularly poor response due to immune 'imprinting' from their previous infection[12,13].

We undertook comprehensive profiling of circulating and mucosal immunity before and after omicron BA.1 or BA.2 infections in a multi-site cohort of triple-vaccinated healthcare workers in the United Kingdom, stratified by those with a history of prior SARS-CoV-2 infection and those who were infection-naive. We find that whilst SARS-CoV-2-specific neutralizing antibody (NAb) responses to omicron infection are indeed lower in those with prior SARS-CoV-2, consistent with recent observations, omicron-specific neutralizing activity nevertheless increases significantly in most individuals. Spike-specific T-cell responses also increase only in those with no prior history of SARS-CoV-2, although the magnitude of these responses is still higher in previously-infected individuals after omicron infection. Furthermore, increases in secretory IgA responses in nasal mucosal lining fluid are seen post-omicron, regardless of prior SARS-CoV-2 history, as are antibody and T-cell responses to non-spike targets. Our data demonstrate that previous SARS-CoV-2 infection history may modulate immune responses to spike upon omicron infection in vaccinated populations. However, omicron infection in triple-vaccinated individuals generally enhances immune responses to SARS-CoV-2 in both blood and mucosa and is likely to contribute to ongoing population immunity against COVID-19.

## Results

### Participants
Ninety-four individuals from four sites (Liverpool, Newcastle, Oxford, and Sheffield) with SARS-CoV-2 infection occurring after three BNT162b2 mRNA vaccine doses were included in the study, of which 38 (40.4%) had a history of prior SARS-CoV-2 before commencing their vaccine course (Table 1). The median time from this first infection to the 3rd vaccine dose was 544 days (IQR 514–559), with all but one infection occurring prior to December 2020 when widespread circulation of B.1.1.7/alpha variants occurred (pre-alpha/ancestral). Previously-infected individuals were slightly older than naive healthcare workers (median age 48 vs. 41, $p = 0.02$, Table 1), with a higher proportion of female participants (84.2% vs. 58.9%, $p = 0.01$). All individuals had received their 1st and 2nd vaccine doses a median of 9.6 weeks apart (IQR 8.9–10.9) in line with UK Health Security Agency (UKHSA) guidelines, which increased the recommended dosing interval from 3 weeks to up to 12 weeks in December 2021[14]. Participants received their 2nd and 3rd vaccine doses a median of 34.9 weeks apart (IQR 32.4–37.4). Omicron infections occurred between 21st December 2021 and 17th May 2022, with sequence data available from 29% of individuals included in the study to confirm SARS-CoV-2 lineage (17 BA.1 and 11 BA.2) (sequence data were obtained as part of UK surveillance, with our access to them covered by participant consent). Assuming that infections from 20th March 2022 were likely to be caused by BA.2[15], 62 (66%) infections were classified as probable or confirmed BA.1 (Table 1). Full details of included participants are provided in Table 1.

### Peripheral blood and nasal immune responses prior to omicron infection
Following the 3rd mRNA vaccine dose but prior to omicron infection, individuals with a history of previous SARS-CoV-2 infection (i.e., hybrid immunity) had significantly higher NAb titers (as assessed by a live-virus neutralization assay) against ancestral ($p = 0.006$), BA.1 ($p = 0.04$), BA.2 ($p = 0.005$) and BA.5 ($p = 0.007$) viruses than SARS-CoV-2 naive healthcare workers (Fig. 1a). While plasma spike-specific IgG was equivalent in both groups, plasma spike-specific IgA was higher in previously-infected individuals to ancestral ($p = 0.002$), BA.2 ($p = 0.01$) and BA.5 ($p = 0.03$) proteins (Fig. 1b, c), as was plasma nucleocapsid-specific IgG ($p < 0.0001$, Fig. 1d). Only 13 of 38 (35.1%) previously-infected individuals had detectable plasma nucleocapsid-specific IgG, likely reflecting waning of anti-N IgG from initial SARS-CoV-2 infection, which occurred almost exclusively in pre-alpha waves. No significant differences between previously-infected and naive participants were seen in spike- or nucleocapsid-specific secretory IgA

**Table 1 | Details of participants included in the study**

|  | SARS-CoV-2 naive | Previously-infected | *p*-value |
|---|---|---|---|
| Total number | 56 | 38 | - |
| Age in years (median with IQR) | 41 (35.0–49.8) | 48 (39.8–55.0) | 0.02 |
| **Sex** |  |  |  |
| Female | 33 (58.9%) | 32 (84.2%) | 0.01 |
| Male | 23 (41.1%) | 6 (15.8%) |  |
| **Omicron subvariant**[a] |  |  |  |
| BA.1 | 39 (8 confirmed) | 23 (9 confirmed) |  |
| BA.2 | 17 (5 confirmed) | 15 (6 confirmed) | - |
| Days from ancestral infection to 3rd mRNA vaccine dose (median with IQR) | - | 544 (514–559) | - |
| Days from 3rd mRNA vaccine dose to omicron infection (median with IQR) | 130 (88–167) | 146 (95–167) | 0.55 |
| Days from 3rd mRNA vaccine dose to pre-omicron sample (median with IQR) | 33 (28–49) | 30 (28–36) | 0.12 |
| Days from omicron infection to post-omicron sample (median with IQR) | 31 (28–34) | 30 (28–36) | 0.92 |
| **Site** |  |  |  |
| Liverpool | 4 | 1 |  |
| Newcastle | 14 | 9 |  |
| Oxford | 22 | 7 |  |
| Sheffield | 16 | 21 | - |

[a]Classified by sequence data or based on a cut-off date of 20th March 2022, when >90% of SARS-CoV-2 infections in the UK were due to BA.2.

*IQR* interquartile range. Comparison of sex between SARS-CoV-2 naive and previously-infected groups was made using Fisher's exact test (two-sided). All other comparisons were made using Mann–Whitney *U*-tests (two-sided). *p*-values not adjusted.

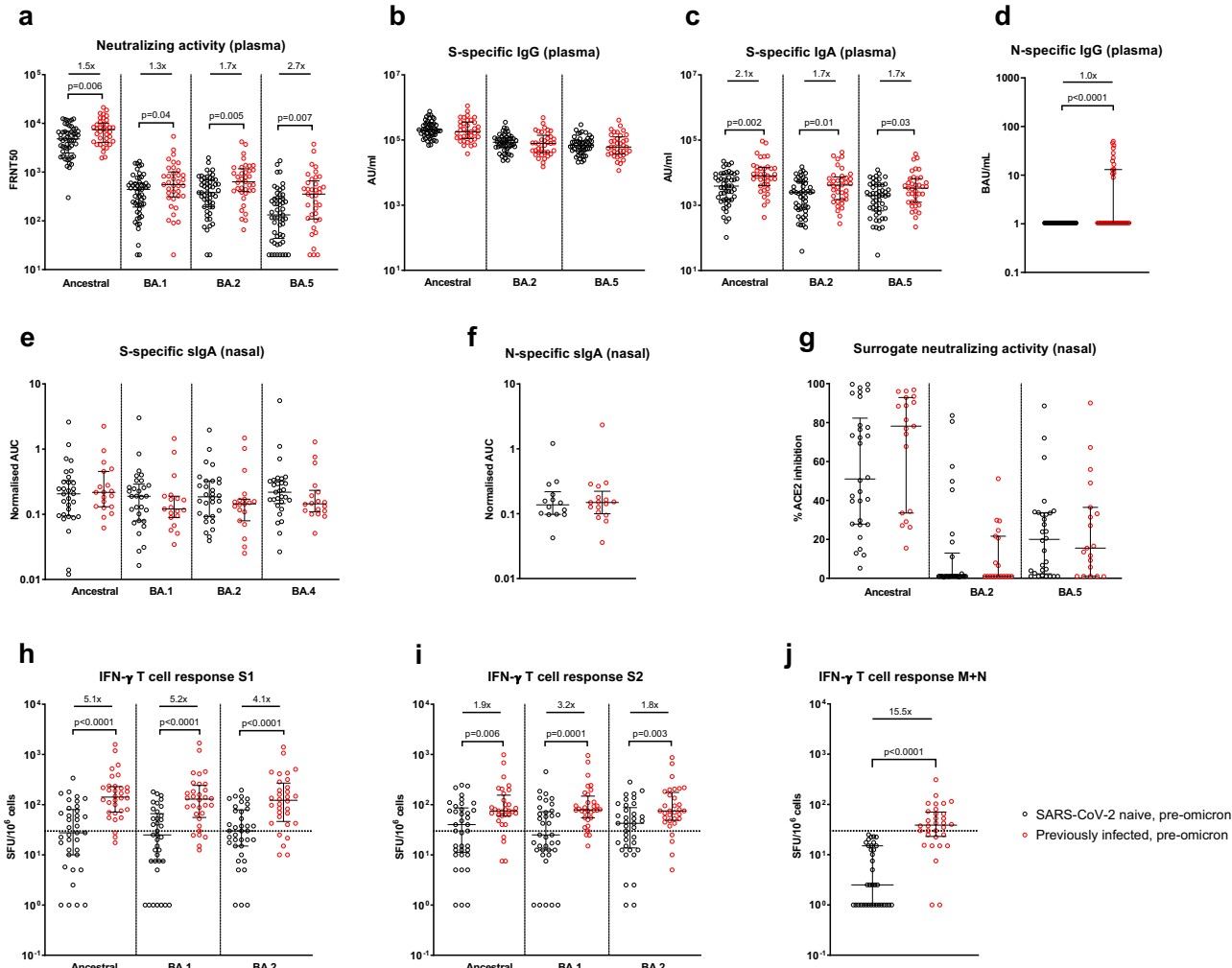

**Fig. 1 | Comparison of immune responses prior to omicron infection in vaccinated SARS-CoV-2-naive and previously-infected individuals.** Samples were taken a median of 32 days (IQR 28-42.3) after 3rd mRNA dose. **a** Live-virus neutralizing activity of plasma against ancestral, BA.1, BA.2, and BA.5 viruses, expressed as the reciprocal of the dilution showing 50% reduction in focus forming units (FRNT50); SARS-CoV-2 spike-specific binding IgG (**b**) and IgA (**c**) in plasma against ancestral, BA.2 and BA.5 spike proteins (AU/mL = arbitrary antibody units/mL in MesoScale Discovery (MSD) assay); **d** nucleocapsid-specific IgG in plasma, assessed by ELISA and expressed in WHO International units, BAU/mL; Secretory IgA (sIgA) in nasal lining fluid targeting (**e**) ancestral, BA.1, BA.2, and BA.4 spike proteins and **f** nucleocapsid-specific sIgA, expressed as area under the curve (AUC) normalized to total sIgA; **g** ability of nasal lining fluid to inhibit ACE2 binding to ancestral, BA.2 and BA.5 spike proteins, assessed by MSD assay; IFN-γ ELISpot responses to overlapping peptide pools representing the S1 (**h**) and S2 (**i**) spike subunits of ancestral,

BA.1 and BA.2 viruses, and **j** a single-pool-containing peptides of both the ancestral membrane (M) and nucleocapsid (N) proteins. Results are expressed as spot-forming units per million cells (SFU/10^6). The dashed line represents a positivity threshold of the mean + 2 SD of the background response. Data are shown with median and interquartile range. Median fold-difference between infection-naive and previously-infected individuals is displayed. All comparisons were made with two-sided Mann–Whitney *U*-test. *p*-values are displayed where <0.05. For **a**–**d**, responses were evaluated in 53 SARS-CoV-2-naive and 37 previously-infected individuals for whom samples were available. For **e**–**g**, responses were evaluated in 32 SARS-CoV-2-naive and 19 previously-infected individuals for whom samples were available. For **h**–**j**, responses were evaluated in 37 SARS-CoV-2-naive and 32 previously-infected individuals for whom samples were available. Source data are provided as a Source Data file.

(sIgA) from nasal lining fluid (Fig. 1e, f), with equivalent human angiotensin-converting enzyme 2 (ACE2) inhibiting activity in both groups against ancestral, BA.2 and BA.5 spike proteins (Fig. 1g). Peripheral IFN-γ T-cell responses were significantly higher in previously-infected individuals against spike S1, S2, and combined membrane and nucleocapsid peptide pools (Fig. 1h–j), as previously demonstrated[9].

## Impact of omicron infection on neutralizing and binding antibody responses

The median time from the 3rd mRNA vaccine dose to omicron infection was 140 days (IQR 90-167, Table 1) and not significantly different between SARS-CoV-2 naive and previously-infected individuals (*p* = 0.55). Following omicron infection, a significant increase in plasma ancestral virus neutralizing ability was seen in previously SARS-CoV-2

naive individuals (2.2-fold, *p* < 0.0001), but not in previously-infected healthcare workers (1.0-fold, *p* = 0.71, Fig. 2a). A much greater increase in neutralizing activity was seen in the previously-naive group to omicron BA.1 (7.3-fold, *p* < 0.0001), BA.2 (5.8-fold, *p* < 0.0001) and BA.5 (8.1-fold, *p* < 0.0001) viruses. Previously-infected individuals also had a boost in NAbs to omicron variants after infection, although to a lesser extent (BA.1, 1.7-fold, *p* = 0.002; BA.2, 1.4-fold, *p* = 0.002; BA.5, 1.6-fold, *p* = 0.007). The post-infection neutralizing titers to all variants tested were significantly higher in the previously-naive individuals than in those with a history of prior SARS-CoV-2 (Fig. 2a). Significant heterogeneity was evident in individual plasma NAb trajectories to BA.1, BA.2, and BA.5 (Fig. 2b), with antibody responders and non-responders seen in both groups. A NAb increase (defined as a fold-difference >1.0) was seen in 49/53 naive individuals to BA.1, 50/53 to BA.2, and 52/53 to

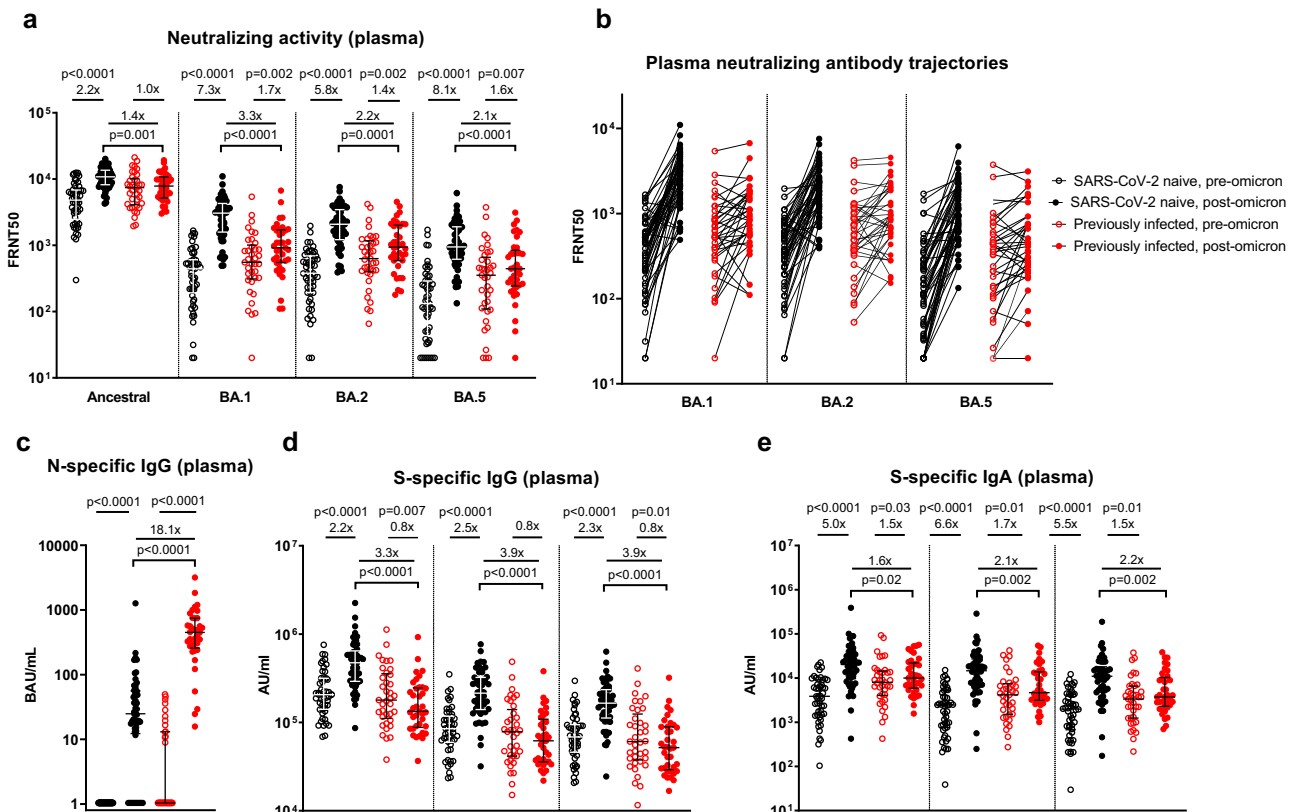

**Fig. 2 | Impact of omicron infection on plasma neutralizing and binding antibodies in vaccinated SARS-CoV-2-naive and previously-infected individuals.** **a** Live-virus neutralizing activity of plasma against ancestral, BA.1, BA.2 and BA.5 viruses, expressed as the reciprocal of the dilution showing 50% reduction in focus forming units (FRNT50); **b** Pair-wise depiction of pre- and post-omicron FRNT as individual participant trajectories for BA.1, BA.2, and BA.5 neutralizing antibodies; **c** Nucleocapsid-specific IgG in plasma, assessed by ELISA and expressed in WHO International units, BAU/mL; SARS-CoV-2 spike-specific binding IgG (**d**) and IgA (**e**) in plasma against ancestral, BA.2 and BA.5 spike proteins (AU/mL = arbitrary antibody units/mL in MSD assay). Data are shown with median and interquartile range. Median fold-change from pre- to post-infection is displayed. Statistical comparisons of paired pre- and post-infection samples made with two-sided Wilcoxon-signed-rank test, and between post-infection levels in previously-infected and SARS-CoV-2 naive individuals made with two-sided Mann–Whitney *U*-test. *P*-values are displayed where <0.05. Responses were evaluated in 53 SARS-CoV-2-naive and 37 previously-infected individuals for whom samples were available. Source data are provided as a Source Data file.

BA.5, compared to NAb increases in 26/37 previously-infected individuals. Fold-change from pre- to post-omicron infection in naive individuals ranged from 0.44× to 29.3× against ancestral SARS-CoV-2, and from 0.51× to 156.4× against omicron subvariants, while in the previously-infected fold-change ranged from 0.50× to 4.9× against ancestral virus, and from 0.19× to 16.4× against omicron subvariants.

Plasma anti-nucleocapsid IgG increased in both groups following infection, although in contrast to the neutralization data, post-infection levels were significantly higher in the previously-infected individuals than in those who were SARS-CoV-2 naive prior to omicron infection (*p* < 0.0001, Fig. 2c). All previously-infected individuals now had detectable anti-nucleocapsid IgG, compared to 41/53 (77.4%) of previously-naive healthcare workers.

Anti-spike binding IgG levels in plasma followed a similar pattern to the NAb data in previously-naive individuals, with an increase following infection seen to ancestral, BA.2 and BA.5 spike proteins, and post-infection levels now significantly higher in these individuals compared to those with prior SARS-CoV-2 (Fig. 2d). In the previously-infected group, anti-spike IgG levels to all proteins were slightly lower following infection compared to their post-dose 3 levels (ancestral, 0.8-fold, *p* = 0.007; BA.2, 0.8-fold, *p* = 0.09; BA.5, 0.8-fold, *p* = 0.01), although this comparison does not account for any waning of responses that occurred after dose 3 but prior to omicron infection. In contrast, plasma anti-spike binding IgA increased following infection in both groups (Fig. 2e), although again to a greater extent

in previously-naive (ancestral, 5.0-fold, *p* < 0.0001; BA.2, 6.6-fold, *p* < 0.0001; BA.5, 5.5-fold, *p* < 0.0001) compared to previously-infected individuals (ancestral, 1.5-fold, *p* = 0.03; BA.2, 1.7-fold, *p* = 0.01; BA.5, 1.5-fold, *p* = 0.01). Similar findings were observed when stratifying participants by likely BA.1 or BA.2 infections (Fig. S1).

**Nasal secretory IgA and functional antibody responses following omicron infection**

Nasal spike-specific sIgA increased significantly following omicron infection in both SARS-CoV-2 naive and previously-infected individuals (Fig. 3a). The extent of increase was similar across the different proteins tested; 11.4-fold to 16.5-fold increases in SARS-CoV-2 naive individuals and 10.6-fold to 14.8-fold increases in previously-infected individuals, with no statistically significant differences in post-infection levels between the two groups. Similarly, nasal nucleocapsid-specific sIgA also increased significantly following omicron infection in both SARS-CoV-2 naive (3.1-fold, *p* = 0.003) and previously-infected (2.9-fold, *p* = 0.0002) healthcare workers (Fig. 3b). The ability of nasal lining fluid to inhibit ACE2 binding to ancestral spike protein (surrogate neutralization assay) was significantly increased post-infection in previously-naive individuals (*p* = 0.003), but not in those with a prior history of SARS-CoV-2 (*p* = 0.95; Fig. 3c). In contrast, significant increases in inhibition of ACE2 binding to both BA.2 and BA.5 spike proteins was seen in both groups (Fig. 3c). Note that

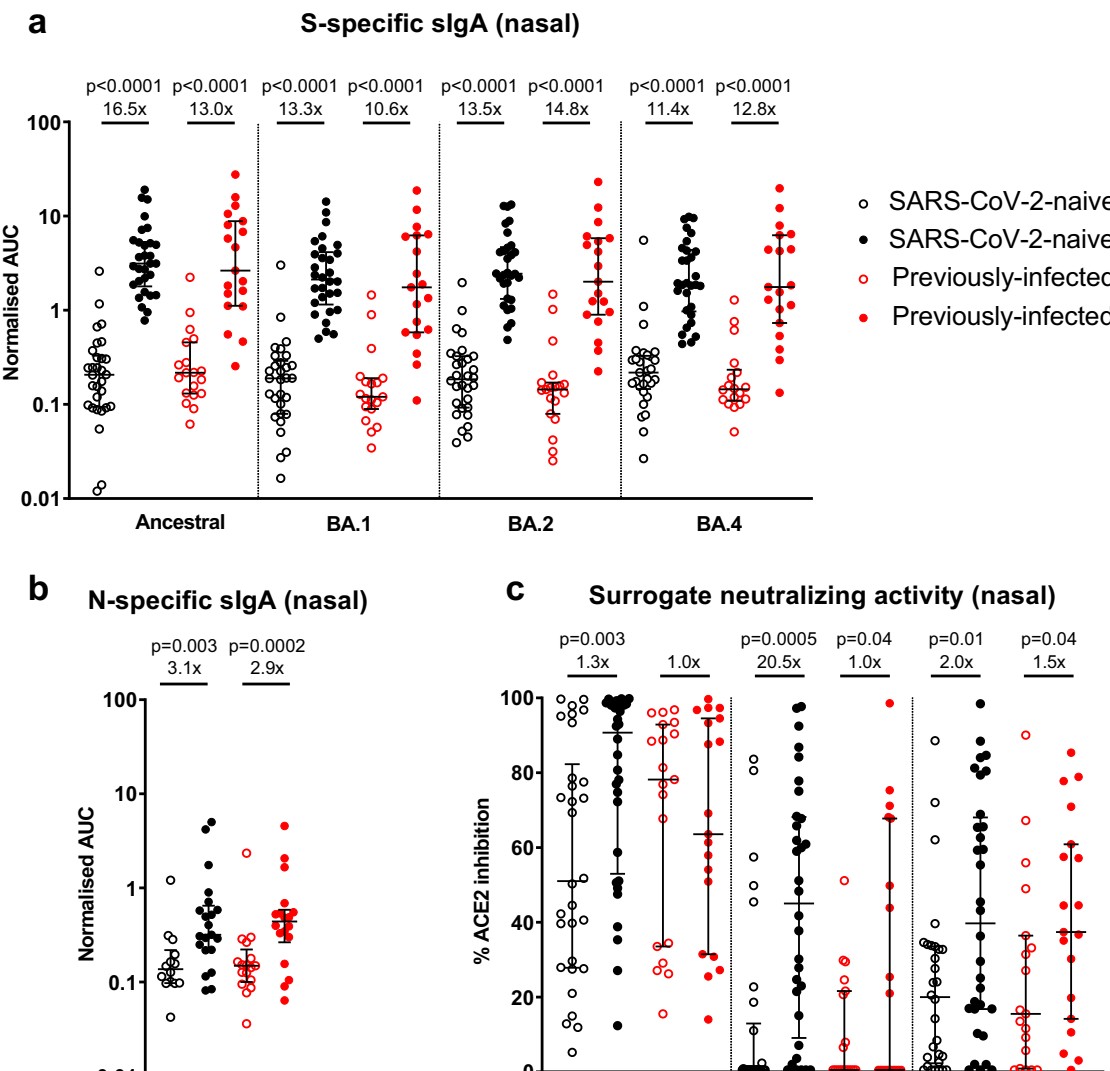

**Fig. 3 | Impact of omicron infection on secretory IgA and ability to inhibit ACE2 binding to spike proteins in nasal lining fluid in vaccinated SARS-CoV-2-naive and previously-infected individuals.** Secretory IgA (sIgA) in nasal lining fluid targeting **a** ancestral, BA.1, BA.2, and BA.4 spike proteins and **b** nucleocapsid protein, expressed as area under the curve (AUC) normalized to total sIgA; **c** ability of nasal lining fluid to inhibit ACE2 binding to ancestral, BA.2 and BA.5 spike proteins, assessed by MSD assay. Data are shown with median and interquartile range.
Median fold-change from pre- to post-infection is displayed. Statistical comparisons of paired pre- and post-infection samples made with two-sided Wilcoxon-signed-rank test, and between post-infection levels in previously-infected and SARS-CoV-2 naive individuals made with two-sided Mann–Whitney U-test. p-values are displayed where <0.05. Responses were evaluated in 32 SARS-CoV-2-naive and 19 previously-infected individuals for whom samples were available. Source data are provided as a Source Data file.

although the fold-change from pre- to post- omicron against BA.2 in the previously-infected group is 1.0×, the p-value = 0.0425; this is due to the use of median fold-change to calculate the statistic.

### Impact of omicron infection on spike and non-spike T-cell responses

In SARS-CoV-2 naive individuals, a significant increase following omicron infection was seen in peripheral IFN-γ ELISpot responses to peptide pools representing ancestral S1 (1.9-fold, $p < 0.0001$) and S2 (1.8-fold, $p = 0.002$), BA.1 S1 (2.7-fold, $p < 0.0001$) and S2 (2.2-fold, $p = 0.0006$), and BA.2 S1 (2.6-fold, $p < 0.0001$) and S2 (1.8-fold, $p = 0.002$; Fig. 4a, b). No increase following omicron was seen in those with a previous SARS-CoV-2 infection, however, in contrast to plasma neutralizing antibodies, post-omicron S1-specific T-cell responses were still significantly higher than in previously-naive individuals for ancestral ($p = 0.003$), BA.1 ($p = 0.004$) and BA.2 ($p = 0.01$)

peptide pools (Fig. 4a). In contrast to spike responses, T-cell responses to a membrane and nucleocapsid peptide pool increased in both SARS-CoV-2 naive (11.8-fold, $p < 0.0001$) and previously-infected (2.0-fold, $p = 0.0004$) individuals, with post-omicron levels significantly higher in the latter group following a boost to previously primed T cells targeting these non-spike antigens ($p = 0.005$, Fig. 4c). Significant increases in ELISpot responses to peptide pools representing ancestral non-structural proteins (NSP)1-2 (2.4-fold, $p = 0.002$), NSP4-6 (2.5-fold, $p = 0.02$), NSP7-11 (1.6-fold, $p = 0.002$) and NSP12 (1.6-fold, $p = 0.04$) were seen in SARS-CoV-2 naive individuals after omicron infection (Fig. S2), although overall responses in each pool were low and often remained below the positivity threshold. Omicron infection induced significant increases to a pool of peptides corresponding to NSP3 amino acids 663-1945 in both naive (2.9-fold, $p = 0.0008$) and previously-infected (2.6-fold, $p = 0.0006$) individuals (Fig. S2).

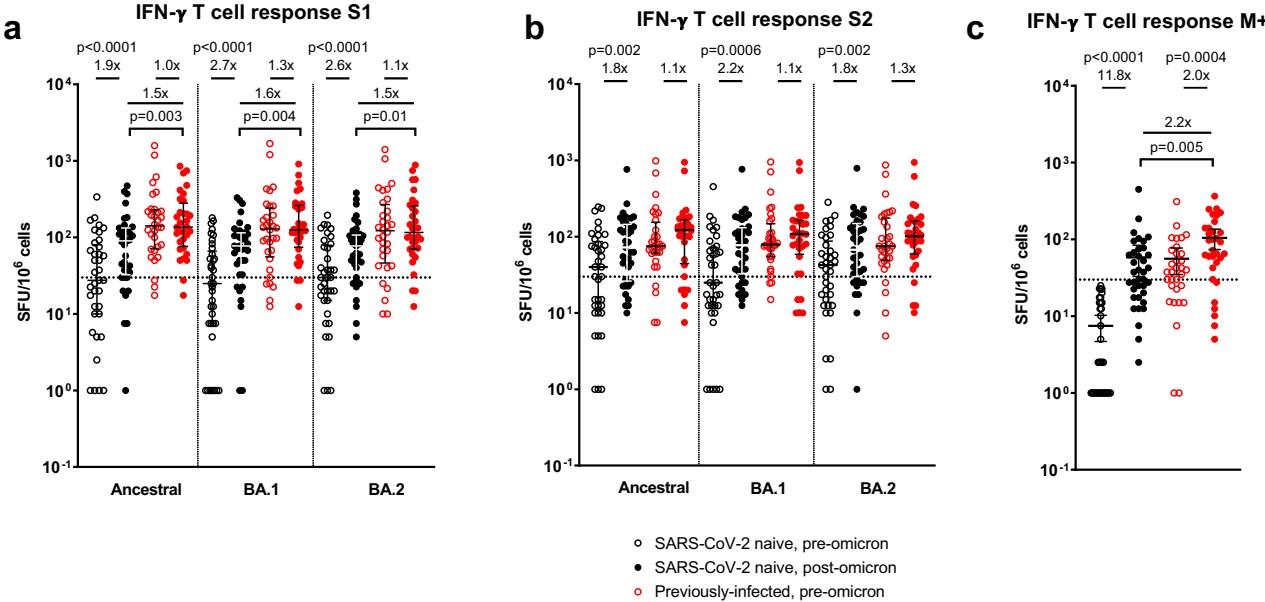

**Fig. 4 | Impact of omicron infection on T-cell response to SARS-CoV-2 proteins in vaccinated SARS-CoV-2-naive and previously-infected individuals.** IFN-γ ELISpot responses to overlapping peptide pools representing the S1 (**a**) and S2 (**b**) spike subunits of ancestral, BA.1 and BA.2 viruses, and **c** a single-pool containing peptides of both the ancestral membrane (M) and nucleocapsid (N) proteins. Results are expressed as spot-forming units per million cells (SFU/10^6). The dashed line represents a positivity threshold of the mean + 2 SD of the background response. Data are shown with median and interquartile range. Median fold-change from pre- to post-infection is displayed. Statistical comparisons of paired pre- and post-infection samples made with two-sided Wilcoxon-signed-rank test, and between post-infection levels in previously-infected and SARS-CoV-2 naive individuals made with two-sided Mann–Whitney U-test. p-values are displayed where <0.05. Responses were evaluated in 37 SARS-CoV-2-naive and 32 previously-infected individuals for whom samples were available. Source data are provided as a Source Data file.

## Evaluation of the heterogeneity in SARS-CoV-2 immune responses before and after omicron infection

A principal component analysis was performed to view the heterogeneity in SARS-CoV-2 immunity when integrating the pre- and post-omicron mucosal and blood immune responses in the 94 individuals included in the study (Fig. 5). 44.6% of the variance observed in the first two dimensions (PC1 and PC2) could be explained by the measured immunological parameters. Omicron infection was a major driver for separation in the data, although there was considerable variability in the impact of omicron across individuals (Fig. 5a). Prior infection status also had a major effect on separation in the data, although, again, considerable overlap was present across individuals (Fig. 5b). Two distinct patterns of immunity were observed, driven either by plasma binding and NAb responses (Fig. 5c, lower right quadrant, highly correlated with dimension 1), or T-cell responses (Fig. 5c, upper right quadrant, highly correlated with dimension 2; Fig. 5d). Plasma antibody and blood T-cell responses were the most important factors in immunophenotypic variability, while variables such as age or time between 3rd vaccine dose and omicron infection contributed very little to the variation in our data (Fig. 5e).

## Impact of omicron infection on spike- and non-spike epitope-specific CD8+ T cells

Given the differences in T-cell responses observed between SARS-CoV-2 naive and previously-infected healthcare workers pre- and post-omicron infection, as well as distinct patterns of response to spike and non-spike proteins, detailed phenotypic characterization of SARS-CoV-2 epitope-specific CD8+ T cells was performed using major histocompatibility complex (MHC) class I-peptide multimer staining and multi-parameter flow cytometry. Immunodominant spike (A*01:01, A*02:01, A*03:01, B*57:01) and non-spike (A*01:01 NSP3, B*07:02 nucleocapsid) epitope-specific CD8+ T cells were characterized, along with cytomegalovirus (CMV)- and Epstein-Barr virus (EBV)-specific

CD8+ populations as controls. In keeping with the IFN-γ ELISpot responses, spike-specific CD8+ T cells increased in magnitude following omicron infection in SARS-CoV-2 naive (p = 0.02), but not previously-infected individuals, whereas non-spike populations increased significantly in both groups (p = 0.002 and p = 0.001; Fig. 6a, b), although in some cases these increases were relatively small. No changes in EBV- or CMV-specific CD8+ T cells were seen.

Epitope-specific T cells were classified into different memory phenotypes as previously described using combinations of CD45RO, CCR7, CD28, and CD95 expression into naive ($T_{NV}$), stem cell memory ($T_{SCM}$), central memory ($T_{CM}$), transitional memory ($T_{TM}$), effector memory ($T_{EM}$), and terminal effector ($T_{TE}$) T cells[16]. After the 3rd mRNA vaccine dose, but prior to omicron infection, spike-specific CD8+ T cells in previously-infected individuals displayed lower $T_{NV}$ phenotypes (p = 0.0027) and higher $T_{EM}$ phenotypes (p = 0.016) than non-spike CD8+ T cells (Fig. S3). Visualization of cell clusters using Uniform Manifold Approximation and Projection (UMAP) plots showed distinct populations of CMV and EBV-specific cells, with some separation of SARS-CoV-2-specific CD8+ T-cell clusters driven by differences in expression levels of several markers, including CD57 and PD-1, which are expressed at higher levels in spike- than in non-spike-specific cells, CD127, which is more highly expressed by non-spike-specific cells, and granzyme B, which is highly expressed in previously-infected but not naive individuals, regardless of epitope (Fig. S4).

The phenotype of spike-specific CD8+ T cells did not change significantly from before to after omicron infection in either previously-naive or -infected individuals (Fig. 7a; S5 and S6). Nevertheless, key differences were seen between the two groups, with spike-specific CD8+ populations in previously-infected individuals displaying fewer $T_{CM}$ and $T_{TM}$ T cells and a more terminal effector phenotype than naive vaccinees, even after omicron infection (p = 0.0017, Fig. 7a), as well as lower levels of CD27 and CCR2 expression than those seen in the naive group (Figs. S5 and S6).

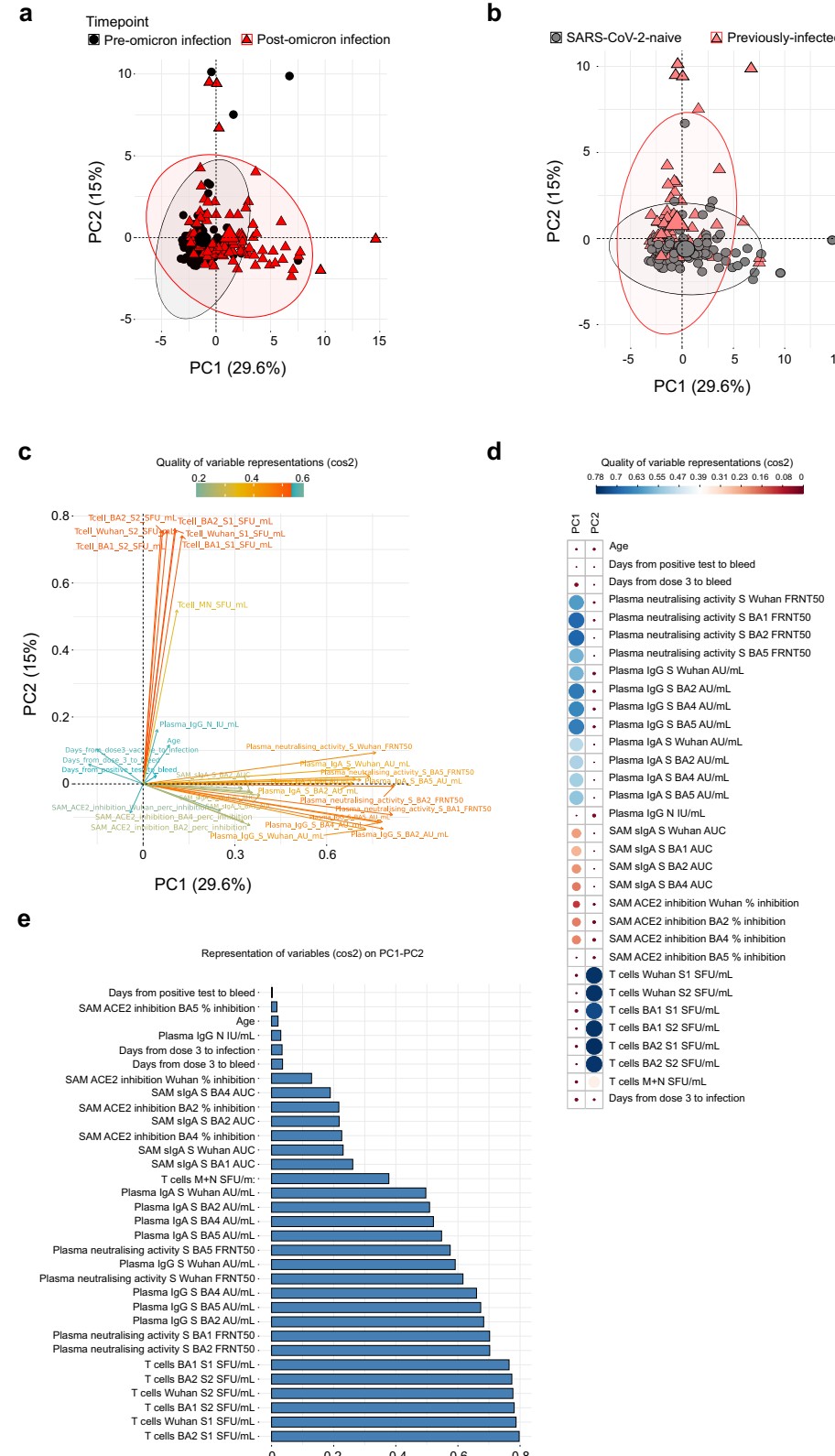

**Fig. 5 | Principal component analysis (PCA) of immune responses before and after SARS-CoV-2 omicron infection in vaccinated SARS-CoV-2-naive and previously-infected individuals.** PCA plot representing integrated immunological data, representing components 1 (PC1) and 2 (PC2) annotated by samples from pre- and post-omicron infection (**a**) and from SARS-CoV-2 naive and previously-infected individuals (**b**). Percentages indicate the variance explained by PC1 and PC2. **c** Variable correlation plot, where positively correlated immune responses are grouped together and negatively correlated variables are found in opposite quadrants. The color indicates the quality of representation of the variable on the principal component (cos2), with higher cos2 equating to greater representation. **d** Quality of variable representations colored by cos2 and contributions of variables to PC1 and PC2 (size of circle, larger circle = greater contribution). **e** All variables included in the PCA, ordered by degree of representation on PC1 and PC2 (cos2).

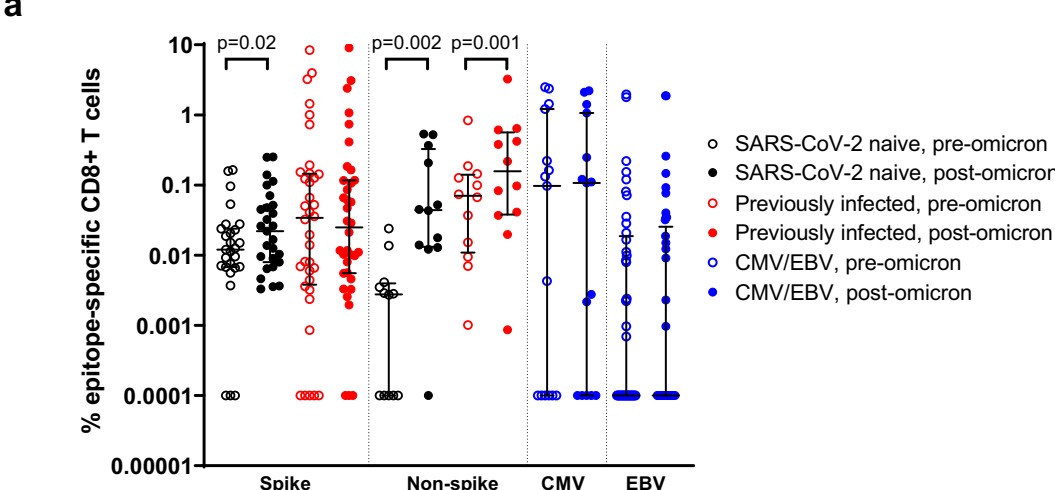

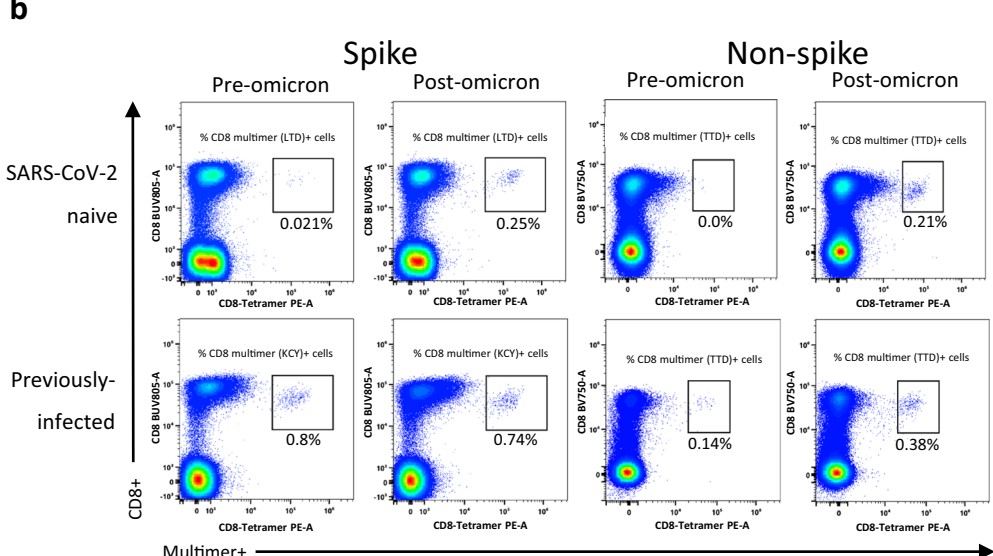

**Fig. 6 | Epitope-specific CD8+ T-cell magnitude before and after omicron infection in vaccinated SARS-CoV-2-naive and previously-infected individuals. a** Magnitude of multimer-specific CD8+ T cells expressed as a % of total CD8+ cells following mRNA vaccine dose 3 (pre-omicron) and after omicron infection (pooled data from Sheffield and Newcastle). Shown are 63 spike-specific multimer populations from 45 individuals (23 naive, 22 previously-infected), 24 non-spike populations from 20 individuals (10 naive, 10 previously-infected), 15 CMV-specific populations from 15 individuals, and 41 EBV-specific populations from 32 individuals. A value of 0.0001% is assigned to negative samples for the purpose of display on a log10 axis. **b** Representative flow cytometry plots showing multimer-specific spike- and non-spike CD8+ populations from naive and previously-infected individuals before and after omicron infection. Data are shown with median and inter-quartile range. Statistical comparisons of paired pre- and post-infection samples where available from the same participants were made with two-sided Wilcoxon-signed-rank test, and between unpaired groups using the Kruskal–Wallis test with Dunn's post hoc test for multiple pairwise comparisons. *p*-values are >0.05 unless displayed. Source data are provided as a Source Data file.

Non-spike CD8+ populations in previously-infected individuals consisted of fewer cells with a $T_{NV}$ phenotype after omicron infection (*p* = 0.023, Fig. 7b). These boosted CD8+ T cells were also more likely to display a terminal effector phenotype than newly primed non-spike populations in the previously-naive group (*p* = 0.040, Fig. 7b), with lower CCR7, but higher granzyme B expression that seen in the naives (Figs. S5 and S6). We also saw differences between the two groups, with CCR2 more highly expressed on CD8+ T cells in naïve participants, while CD57 and CD38 were more highly expressed in those with a history of previous infection (Figs. S5 and S6). Expression of the cytolytic enzyme granzyme B in spike-specific CD8+ T cells did not change significantly after omicron infection, however, expression levels were significantly higher in previously-infected individuals at both time points (*p* = 0.0002 and *p* = 0.0042, Fig. 7c), to a level similar to that seen in CMV-specific T cells.

## Discussion

In countries with high SARS-CoV-2 vaccination rates, the emergence of omicron-lineage viruses with considerable immune escape from vaccine-induced NAbs has led to high levels of infection followed by hybrid immunity. The nature of this post-omicron immunity has been questioned in two main ways by recent findings[12,13]. Firstly, it has been suggested that the relatively attenuated nature of omicron infections[11] results in poor induction of immune responses, and secondly that 'imprinting' from previous SARS-CoV-2 infection (e.g., with ancestral or alpha viruses) leads to profound immune-dampening; with both scenarios precluding the generation of omicron-specific immunity able to protect against future omicron infections. These concerns were raised on the basis of data focused primarily on circulating NAbs, mediated by spike-specific responses. We have taken a broader approach and characterized mucosal and circulating immunity to spike and non-

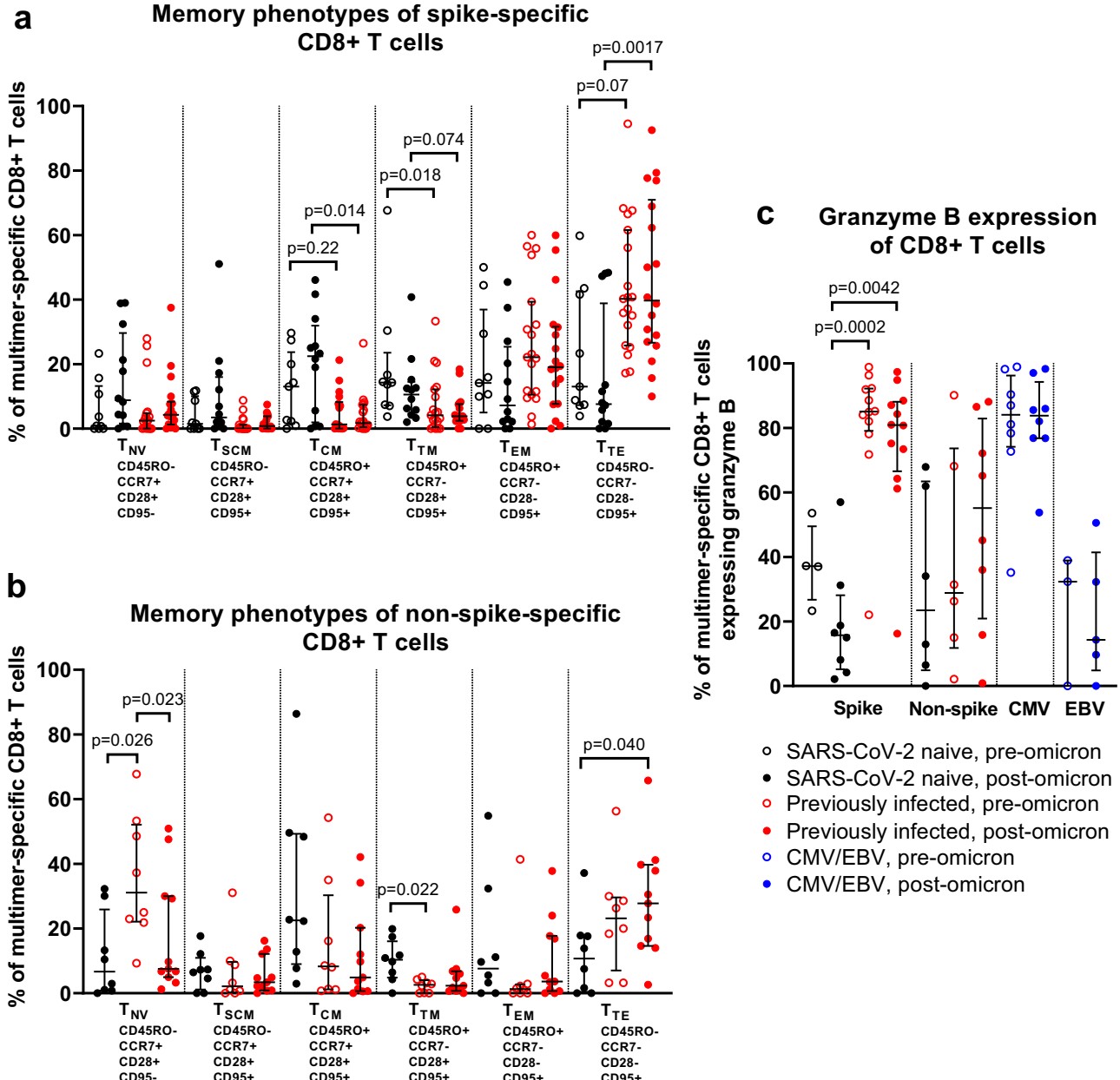

**Fig. 7 | Epitope-specific CD8+ T-cell phenotypes before and after omicron infection in vaccinated SARS-CoV-2-naive and previously-infected individuals.** **a** Memory phenotypes of spike-specific CD8+ T cells in 11 SARS-CoV-2 naive and 14 previously-infected individuals (pooled data from Sheffield and Newcastle). **b** Memory phenotypes of non-spike CD8+ T cells in 7 SARS-CoV-2 naive and 10 previously-infected individuals (pooled data from Sheffield and Newcastle). **c** Granzyme B expression in epitope-specific CD8+ T cells (data from Sheffield). Shown are spike-specific populations from 8 naive and 12 previously-infected individuals, non-spike populations from 6 naive and 8 previously-infected

individuals, and CMV- and EBV-specific populations from 8 and 5 individuals, respectively. Data are shown with median and interquartile range. Statistical comparisons of paired pre- and post-infection samples where available from the same participants were made with two-sided Wilcoxon-signed-rank test, and between unpaired groups using the Kruskal–Wallis test with Dunn's post hoc test for multiple pairwise comparisons. *p*-values are >0.05 unless displayed. $T_{NV}$ naive T cells, $T_{SCM}$ stem cell memory T cells, $T_{CM}$ central memory T cells, $T_{TM}$ transitional memory T cells, $T_{EM}$ effector memory T cells, $T_{TE}$ terminal effector T cells. Source data are provided as a Source Data file.

spike antigenic targets. While we replicate some of the previously reported NAb data, we also demonstrate that omicron infections result in significant increases in mucosal antibodies and circulating T cells, with the dominant antigenic targets of some responses dependent on prior SARS-CoV-2 infection history.

We and others have repeatedly shown that individuals with a history of SARS-CoV-2 prior to their 1st mRNA vaccine have higher NAb titers after each SARS-CoV-2 vaccine dose when compared to infection-naive individuals[8–10]. This advantage is eliminated immediately after omicron infection in triple-vaccinated adults, with

omicron-specific NAbs getting a greater boost in previously-naive healthcare workers. This unexpected finding in triple-vaccinated individuals was first reported by Reynolds et al.[13], who noted that post-omicron NAb titers in 11 previously-naive healthcare workers were significantly higher than in 6 individuals with a history of infection with pre-alpha SARS-CoV-2 viruses. Our larger cohort confirms this finding, but also demonstrates a great deal more heterogeneity, with 70% of previously-infected individuals increasing omicron-specific NAb titers in response to BA.1 or BA.2 infection. A study of 56 triple-vaccinated healthcare workers in Sweden also demonstrated that post-omicron

IgG and NAb levels were higher in previously-naive individuals compared to those with a history of prior SARS-CoV-2 infection[12]. Importantly, omicron-specific antibody immunity was still higher in both groups compared to healthcare workers who did not have breakthrough omicron. Several other studies have confirmed that omicron infections in previously-naive triple-vaccinated adults increase omicron-specific NAb activity[17], primarily through the expansion of memory B cells against epitopes that are conserved across variants.

The mechanism behind the relatively poor induction of circulating NAbs by omicron infection in triple-vaccinated previously-infected individuals remains unexplained. Imprinting or original antigenic sin (OAS) from prior infections has been suggested as a possible explanation[13]. The concept of OAS was first described for influenza antibodies[18], with imprinting from prior influenza infections on immune responses following subsequent influenza infection and vaccination noted in more recent work[19–21]. OAS from antibodies generated by prior OC43/HKU1 seasonal coronavirus infections occurs; and is seen more potently following SARS-CoV-2 infection than after the 1st mRNA vaccine dose, with SARS-CoV-2 infection driving a boost of lower affinity antibodies[22]. Potentially beneficial imprinting has also been described; omicron breakthrough infections in infection-naive vaccinated individuals induce antibodies from B cells that cross-react with receptor-binding domains from multiple variants and provide a greater breadth of protective immunity whereas antibodies following primary omicron infections in infection-naive unvaccinated individuals are of much narrower specificity[17]. Of note, we did not observe a boosting of NAbs against an ancestral strain of SARS-CoV-2 at the expense of generating omicron-specific immunity. Importantly, even individuals with prior infection had a much greater increase in BA.1/BA.2/BA.5 NAb activity than to the ancestral strain. While it is likely that conserved epitopes between ancestral and omicron viruses would have been preferentially boosted over omicron-specific epitopes, it is unlikely that OAS alone fully explains the attenuated response in previously-infected individuals we observed.

Another hypothesis is that the higher omicron-specific immunity after the 3rd mRNA vaccine dose seen in those with prior SARS-CoV-2 infection may lead to lower viral loads and therefore reduced antigenic exposure during omicron infection, resulting in muted immune induction. While Blom et al.[12] did find a correlation between cycle threshold (Ct) values and NAb levels, nadir Ct values (i.e., highest viral loads) were no different between naive and previously-infected healthcare workers. The amount of virus in previously-infected individuals in our study was certainly sufficient to generate significant increases in non-spike antibodies and T cells; however, this does not rule out the possibility that antigenic load was lower in this group, impacting the post-omicron immunity observed. Finally, it is possible that differences in spike-specific B cell phenotype could underlie the variability in NAb responsiveness to omicron infection. A link between the frequency of CD27lo B cells and altered B cell receptor (BCR) signaling with the response to the 3rd mRNA vaccine dose has been reported[23]. Although in that particular study, recent infection prior to vaccination drove a B cell phenotype that led to muted vaccine response, future work should explore whether molecular mechanisms that impair BCR signaling are more prevalent in previously-infected individuals after a 3rd mRNA vaccine dose than in naive triple-vaccinated adults.

We found that spike-specific mucosal antibodies were low after the 3rd mRNA vaccine dose and similar between SARS-CoV-2 naive and previously-infected individuals. We and others have shown that mucosal IgA is induced by mRNA vaccines in those with a history of prior SARS-CoV-2[24,25], so this finding was unexpected and is perhaps explained by our selection of a cohort of healthcare workers who experienced an omicron infection, thereby potentially enriching for previously-infected individuals at increased risk of breakthrough infection due to low mucosal immunity. We have also previously noted

a waning of mucosal IgA over 9 months after SARS-CoV-2 infection[26] and the time to first infection in our previously-infected cohort was much longer. Nevertheless, omicron infection induced mucosal SARS-CoV-2-specific secretory IgA and omicron-specific NAb responses, regardless of prior SARS-CoV-2 infection history. Mucosal immunity is likely to be important both for protection from infection and transmission-blocking immunity, yet it is poorly induced by currently licensed parenteral vaccines. Our findings and those from others[17,27] suggest that omicron breakthrough infections in vaccinated individuals may contribute to filling this immunity gap.

We demonstrate that omicron infection increased spike-specific T-cell responses in previously-naive individuals, and non-spike-specific T cells regardless of prior SARS-CoV-2 history, with membrane and nucleocapsid-specific T cells boosted to a greater degree in previously-infected healthcare workers. These findings are in contrast to those in Blom et al.[12], who found no increase in spike-specific responses after omicron infection using the Oxford Immunotech (OI) T-spot assay, and similar M- and N-specific T-cell responses in both previously-infected and -naive individuals post-omicron. These differences may be explained by the assays used, as we have previously reported higher magnitude responses in our ELISpot assay when compared to the OI assay, which is not marketed as a quantitative test[28]. In our study, despite not increasing spike-specific T cells following omicron infection, previously-infected individuals still maintained higher S1-specific responses than their previously-naive counterparts.

Characterization of immunodominant CD8+ spike epitope-specific cells in previously-infected individuals also demonstrated a more terminally differentiated, highly cytotoxic phenotype. The high granzyme B content in these T cells was strikingly similar to that seen in CMV-specific CD8+ T cells, shown previously to have a late-differentiated phenotype with high cytotoxic potential distinct from other anti-viral CD8+ T-cells[29]. Our findings suggest that an infection prime followed by 3 mRNA vaccine doses results in maximally induced spike-specific T-cell responses, with limited potential for further boosting, at least in the short term. Interestingly, the granzyme B content of spike-specific CD8+ T cells in previously-naive individuals after omicron infection was much lower than in previously-infected individuals prior to omicron. As both these scenarios represent immunity after four spike exposures, the order of exposures may matter in the generation of hybrid immunity with respect to the cytotoxic potential of CD8+ T cells. T-cell responses to non-spike targets increased significantly in all individuals following omicron infection, with those with prior SARS-CoV-2 having higher post-omicron levels. A similar advantage in nucleocapsid-specific IgG was observed in previously-infected individuals following omicron infection. While these antibodies would not be expected to contribute to NAb activity, a protective role through antibody-dependent cellular cytotoxicity or other non-neutralizing effector functions may exist[30,31]. While 22.6% of naive participants did not mount a detectable N-specific IgG response upon omicron infection, this is consistent with UKHSA data showing that breakthrough infections in fully vaccinated individuals result in poor induction of N-specific IgG, possibly due to the lower disease severity experienced[32].

The key question is whether omicron infections in vaccinated individuals confer a degree of protection against subsequent omicron-lineage viruses. Real-world effectiveness data are aligned with our findings that omicron infections induce or boost diverse immune responses in vaccinated individuals. In Portugal, where 98% of the study population had completed a primary 2-dose vaccine course prior to 2022, and mRNA vaccine booster coverage was 82% at the start of the BA.5 wave, the protective efficacy of BA.1/BA.2 breakthrough infections against subsequent BA.5 infection was 75.3% compared to infection-naive vaccinees, whereas previously efficacy following breakthrough infections with other variants was lower (e.g., 54.8% following alpha, 61.3% following delta)[33]. Similarly, prior omicron

infection in triple-vaccinated individuals was found to be 93.6% protective against subsequent BA.5 infection in a Danish cohort study[34].

Our study has several limitations which need to be considered and may affect the generalizability of our findings. Most importantly, we selected previously-infected individuals who had breakthrough omicron infections within a few months of a 3rd mRNA vaccine dose, and who therefore may not be immunologically similar to those who remained protected for longer. Real-world protection from symptomatic BA.1/BA.2 infection is greater in those with infection followed by three mRNA vaccine doses compared to SARS-CoV-2 naive triple-vaccinated individuals[35], therefore we likely enriched for previously-infected individuals with lower protective immunity. In most experiments, we also deliberately considered responses to whole proteins rather than specific epitopes to capture the breadth of immunity generated through SARS-CoV-2 infection. It is likely that individual epitope-specific differences may exist between groups, including those affected by mutations in omicron-lineage viruses. Additionally, our cohort of healthcare workers is relatively young and healthy, and so not entirely representative of the general population.

Although vaccination remains the safest way to acquire SARS-CoV-2 immunity, accumulating data suggest that infection-induced immunity in SARS-CoV-2 vaccinees can provide enhanced protection. For example, protection against BA.4/5 infection is estimated to be more durable in UK adults with a delta/omicron breakthrough infection after two vaccine doses, compared with triple-vaccinated adults with no history of infection[36]. To date, there is epidemiological evidence that hybrid immunity gives good protection against BA.4/5, for example, protection against hospitalization with BA.4/5 was greater than 90% for at least 6-8 months in a Quebec study of adults over 60 years of age who had received at least two vaccine doses and one infection at any time[37]. As highly vaccinated populations experience increasing numbers of SARS-CoV-2 infections, the nature of hybrid immunity in individuals becomes more complex and heterogeneous. We have demonstrated that immune components not induced by currently available vaccines such as mucosal and non-spike responses are enhanced by viral infection. These may play a critical role in accumulating protective immunity to SARS-CoV-2[38], but should also be prioritized as targets to broaden the responses generated by the next generation of vaccines.

## Methods

### Participant recruitment

The PITCH (Protective Immunity from T cells to Covid-19 in Health workers) study is a prospective observational cohort study of 2149 healthcare workers (HCWs) recruited at five sites in the UK (University Hospitals Birmingham NHS Foundation Trust, Liverpool University Hospitals NHS Foundation Trust, Newcastle upon Tyne Hospitals NHS Foundation Trust, Oxford University Hospitals NHS Foundation Trust, and Sheffield Teaching Hospitals NHS Foundation Trust). Eligible participants were adults aged 18 years and over, currently working as healthcare workers, including allied support and laboratory staff. Individuals were recruited by word of mouth, hospital email communications, from hospital-based staff SARS-CoV-2 screening programs, as well as enrolling through the wider SIREN study, of which PITCH is a sub-study. The SIREN study is registered with ISRCTN (Trial ID:252 ISRCTN11041050), and was approved by the Berkshire Research Ethics Committee, Health Research 250 Authority (IRAS ID 284460, REC reference 20/SC/0230), with PITCH recognized as a sub-study on 2 December 2020. Some participants were recruited under other protocol-aligned REC-approved studies; in Liverpool, some participants were recruited under the "Human immune responses to acute virus infections" Study (16/NW/0170), approved by North West-Liverpool Central Research Ethics Committee on 8 March 2016, and amended on 14th September 2020 and 4th May 2021. In Oxford, participants were recruited under the GI Biobank Study 16/YH/0247,

approved by the research ethics committee (REC) at Yorkshire & The Humber - Sheffield Research Ethics Committee on 29 July 2016, which has been amended for this purpose on 8 June 2020. In Sheffield, participants were recruited under the Observational Biobanking study STHObs (18/YH/0441), which was amended for this study on 10 September 2020. All procedures were conducted in accordance with the principles of the Declaration of Helsinki (2008), and the International Conference on Harmonization Good Clinical Practice guidelines. All participants enrolled provided written informed consent.

Participants were defined as either SARS-CoV-2 naive or previously-infected at the time of enrollment in the PITCH study based on documented PCR and/or serology from local NHS trusts, or from MSD analysis of S and N plasma IgG levels of PITCH samples[8]. Patients were sampled 28 days following their third vaccine dose, and 28 days following documented SARS-CoV-2 infection. At each visit synthetic absorption matrix (SAM) strips were collected to obtain nasal lining fluid and heparinized whole blood was collected, which was separated via density gradient centrifugation into plasma and peripheral blood mononuclear cells (PMBCs). Nasal lining fluid was sampled by inserting the SAM strip into the nostril and holding it against the mucosa for 1 min via light finger pressure on the outside of the nose.

### NAb responses

The neutralizing ability of plasma was measured using a Focus Reduction Neutralization Test (FRNT). Briefly, serially diluted plasma was mixed with live SARS-CoV-2 virus of an ancestral strain (Australia/VIC01/2020), BA.1, BA.2, or BA.5, then incubated at 37 °C for 1 h. The plasma-virus mixture was then transferred to 96-well, cell culture-treated, flat-bottom microplates with confluent monolayers of Vero cells (ATCC, CCL-81) in duplicate and incubated at 37 °C for a further 2 h. After incubation, 1.5% carboxymethyl cellulose (CMC) overlay medium diluted in Dulbecco's Modified Eagle Medium (Gibco, 31966047) with 1% FBS (DMEM1) was added into each well. A focus-forming assay was performed after cells had been incubated with the virus at 37 °C for 24 h. Vero cells were fixed with 4% Formaldehyde (Sigma, F8775), permeabilized with 2% Triton-X (Sigma, T9284), then stained with human anti-N mAb (mAb206, produced in-house[39]) at 2 μg/mL followed by peroxidase-conjugated goat anti-human IgG (Sigma, A0170). After staining, the foci were visualized by adding TrueBlue Peroxidase Substrate (SeraCare, 5510-0030), and approximately 100 foci were observed in the wells without plasma. Infection plates were counted on the AID EliSpot reader using AID ELISpot software. The percentage of focus reduction was calculated for each plasma and FRNT50, the reciprocal dilution of plasma required to neutralize 50% of input virus, was determined using the probit program from the SPSS package.

### Spike-specific IgG- and IgA-binding antibody responses

SARS-CoV-2-specific IgG and IgA binding antibody levels in plasma were assessed using the multiplex Mesoscale Discovery (MSD) platform V-PLEX SARS-CoV-2 Panel 27 IgG (Meso Scale Diagnostics, K15606U) and V-PLEX SARS-CoV-2 Panel 27 IgA (Meso Scale Diagnostics, K15608U) kits. Assays were performed as per the manufacturer-recommended protocol. Plates pre-coated with SARS-CoV-2 spike antigen spots (including Wuhan-Hu-1, and omicron sublineages BA.2, and BA.5) were blocked with 150 μL Blocker A solution for 30 min at room temperature (RT), shaking at 800 rpm. No BA.1 sublineage protein was present on kit 27 plates. Plates were washed, and 50 μL/well plasma samples diluted to 1:40000 in Diluent 100 were loaded in duplicate and incubated for 2 h at RT, shaking at 800 rpm. Plates were washed, and 50 μL/well of 1× detection antibody solution was added to each well and incubated for 1 h, shaking at 800 rpm. Plates were washed and 150 μL MSD GOLD Read Buffer B was added to wells, before reading immediately with a MESO® SECTOR S 600 instrument (Discovery Bench software version 4.0).

## Nucleocapsid IgG antibody detection

Nucleocapsid IgG was assessed using an in-house ELISA[40]. High-binding 96-well ELISA plates (Immulon 4HBX; Thermo Scientific, 6405) were coated overnight at 4 °C with 50 μL/well full-length untagged nucleocapsid protein produced in *Escherichia coli* (Uniprot ID P0DTC9 (NCAP_SARS2)), diluted to 2 μg/mL in 7.4 pH phosphate buffered saline (PBS). Plates were washed with 0.05% PBS-Tween, then blocked for 1 h with 200 μL/well 0.5% casein buffer. Plasma samples were diluted to 1:200, and 100 μL loaded in duplicate wells. Plates were incubated for 1 h at RT, then washed and loaded with 100 μL/well of goat anti-human IgG-HRP conjugate (Invitrogen, 62-8420) at 1:500. Plates were incubated for 2 h at RT, then washed and developed for 10 min with 100 μL/well TMB substrate (KPL, 5120-0074) and stopped with 100 μL/well HCl Stop solution (KPL, 5150-0021). Absorbance at 450 nm (A450) was read immediately with a HIDEX sense luminometer (HIDEX sense plate reader software version 1.2.1.).

To allow quantification of antibody concentration, we included a 12-step standard curve consisting of sera pooled from convalescent SARS-CoV-2-confirmed patients, calibrated to the WHO International Standard for anti-SARS-CoV-2 immunoglobulin (NIBSC, 20/136), with results reported in binding antibody units/mL (BAU/mL). As previously reported[40], the serostatus of samples was determined based on a threshold selected to maximize sensitivity, validated using sera from PCR-confirmed SARS-CoV-2 convalescent patients and pre-2019 samples. Samples considered negative according to this threshold were assigned a value of 1.04 BAU/mL, half the value of the lowest point on the standard curve.

## Secretory IgA

Levels of spike- and nucleocapsid-specific dimeric secretory IgA (sIgA) present in nasal lining fluid were assessed in an ELISA using a primary antibody targeting the human secretory component. Levels of total sIgA were also assessed to allow normalization of SARS-CoV-2-specific sIgA results.

To detect SARS-CoV-2 antibodies, high-binding 96-well ELISA plates (Immulon 4HBX; Thermo Scientific, 6405) were coated overnight at 4 °C with 50 μL/well of SARS-CoV-2 spike proteins representing the SARS-CoV-2 D614G (Sino Biological 40589-V08H8), omicron BA.1 (40589-V08H26), BA.2 (40589-V08H28) or BA.4 (40589-V08H32) viruses, diluted to 1 μg/mL in 7.4 pH PBS, or full-length untagged nucleocapsid protein produced in *Escherichia coli* (Uniprot ID P0DTC9 (NCAP_SARS2)), diluted to 2 μg/mL in 7.4 pH PBS. For total sIgA ELISAs, plates were coated overnight with goat anti-human kappa and lambda light chain antibodies (Southern Biotech, 2060-01, 2070-01) each diluted to 2 μg/mL in Dulbecco's phosphate-buffered saline (DPBS). Plates were washed with 0.05% PBS-Tween and blocked for 1 h with 200 μL/well 1% casein buffer.

For detection of SARS-CoV-2-specific sIgA, samples were tested in duplicate in a 5-well dilution series proceeding in 2x steps from 1:10 to 1:160. Plates were loaded with 50 μL/well of sample and incubated overnight at 4 °C. A curve was generated for each sample by plotting $A_{450}$ against a dilution coefficient. This was used to calculate an area under the curve (AUC) value for each sample. To detect total sIgA, nasal lining fluid was tested in duplicate wells at 1:4000. For quantification purposes a standard curve of human IgA from colostrum (Sigma, I2636) was included in a 12-well dilution series proceeding in 2× steps from an initial dilution of 1 μg/mL. Plates were incubated for 1 h at 36 °C. Following sample incubation, plates (both specific and total) were washed and loaded with 100 μL/well mouse anti-human secretory component antibody (HP6141, Calbiochem, 411423) at 1 μg/mL for 2 h at RT. Plates were washed and loaded with goat anti-mouse IgG-HRP conjugate (Invitrogen, 31439) at 1:500 for 1 h at RT, then washed and developed for 5 min with 50 μL/well TMB substrate (KPL, 5120-0074) before addition of 50 μL/well HCl stop solution (KPL, 5150-0021). A450 was read immediately with a HIDEX sense luminometer.

SARS-CoV-2-specific sIgA AUC values were normalized to total sIgA from the same sample.

## Surrogate neutralizing activity in nasal lining fluid

SARS-CoV-2-specific surrogate neutralizing ability of nasal lining fluid was assessed using the MSD V-PLEX SARS-CoV-2 Panel 27 ACE2 kit (Meso Scale Diagnostics, K15609U), which assesses the ability of samples to prevent binding of ACE2 to SARS-CoV-2 spike antigens, including Wuhan-Hu-1, and omicron sublineages BA.2, and BA.5. SAM strips were eluted into a buffer of PBS 1% BSA, and protease inhibitor cocktail I (Calbiochem, 539131). Plates pre-coated with SARS-CoV-2 antigen spots were blocked with 150 μL Blocker A solution for 30 min at RT, shaking at 800 rpm. Plates were washed, and eluted nasal lining fluid samples were loaded neat at 25 μL/well in duplicate and incubated for 1 h at RT, shaking at 800 rpm. After incubation, 25 μL of 1× SULFO-TAG Human ACE2 Protein solution was added to wells without washing or aspiration of the sample, and incubated for 1 h at RT, shaking at 800 rpm. Plates were washed, and 150 μL/well MSD GOLD Read Buffer B was added, before reading immediately with a MESO® SECTOR S 600 instrument. The neutralizing activity of samples was expressed as the percentage of ACE2 inhibition compared to a condition with no nasal lining fluid on the same plate.

## IFN-γ ELISpot assays

IFN-γ ELISpot assays were performed on cryopreserved peripheral blood mononuclear cells (PBMCs) using the ELISpot Flex Human IFN-γ kit (Mabtech, 3420-2 A). Assays were performed in Sheffield, Oxford, and Newcastle, using the same previously harmonized and published protocol[8]. Overlapping peptide pools (18-mer peptides with 10 amino acid overlap, Mimotopes) representing the S1 and S2 subunits of SARS-CoV-2 Wuhan-hu-1, omicron BA.1 and omicron BA.2 spike proteins, pooled ancestral Wuhan-hu-1 membrane and nucleocapsid proteins, and non-structural proteins (NSP1+2, NSP3b+c, NSP4-6, NSP7-11, NSP12, and NSP13+14) were used to stimulate PBMCs at 2 μg/mL. Pools of CMV, EBV and influenza peptides (CEF) at 2 μg/mL and phytohemagglutinin (PHA) at 1 μg/mL were also included as positive controls, as well as a cell-only condition. Sterile 96-well plates with 0.45 μm PVDF membrane were coated with 50 μL/well of anti-IFN-γ coating antibody diluted to 10 μg/mL in sterile PBS, and incubated overnight at 4 °C. Plates were washed with sterile PBS and blocked with 200 μL/well R10 (RPMI + 10% FBS + 1% pen/strep). After at least 2 h incubation at 37 °C, wells were emptied, and 200,000 cells diluted in 50 μL R10 were added to each well. 50 μL of peptide pools were added to wells, and plates were incubated for 18–24 h at 37 °C, washed, and 100 μL/well of biotinylated secondary antibody diluted to 1 μg/mL in PBS 0.5% BSA loaded before incubation for 2–4 h at RT. Plates were washed and 100 μL/well of streptavidin-ALP conjugate diluted in PBS to 1 mg/mL was added to wells and incubated for 40 min at RT. Plates were developed using the AP conjugate substrate kit (Biorad, 170-6432). Plates were washed and 100 μL of detection solution was added to wells and incubated for 15 min at RT. Plates were washed with tap water and left to dry before reading on the AID EliSpot reader. Spots were counted using AID ELISpot software version 8.0, duplicate wells were averaged and cell-only value subtracted, and virus-specific responses were expressed as spot-forming units (SFUs)/10^6 cells. A cut-off for positivity was determined by taking the mean + 2 SD of the SFU/10^6 cell value of all cell-only control wells. Negative results were assigned a value of 1 SFU/10^6.

## DNA extraction and HLA typing

DNA was isolated from cryopreserved whole blood samples using the DNeasy® Blood and Tissue kit (QIAGEN, 69504) as per the manufacturer's protocol. Newcastle-based participant HLA typing for Class I (HLA-A; B; C) was performed by MC Diagnostics Limited (Wales, United Kingdom). Sheffield participant HLA Class I typing was performed at

The University of Oxford by amplifying Exons 2 and 3 for each locus using in-house sequence-specific primers and sequenced on an Applied Biosystems AB3730 instrument. Analysis was done by comparing heterozygote traces to known sequences on the IMGT-HLA database to 4-digit resolution[41].

## Multi-parameter flow cytometry characterizing antigen-specific CD8+ T cells

Cryopreserved PBMCs were rapidly thawed and rested for 2–4 h in R10 media (RPMI + 10% FBS + 1% pen/strep) at 37 °C. After resting and prior to staining, cells were washed in PBS and divided into FACS tubes at $2-3 \times 10^6$ cells per sample. Firstly, samples and HLA mismatched negative control donors were stained with PE-conjugated SARS-CoV-2 spike or non-spike-specific MHC pentamer/dextramers (PE-HLA-A*03:01 KCYGVSPTK $S_{378}$, ProImmune, peptide code 4443, PE-HLA-A*02:01 YLQPRTFLL $S_{269}$, PE-HLA-A*01:01 LTDEMIAQY $S_{865}$, PE-HLA-B*57:01 GTITSGWTF $S_{879}$, PE-HLA-A*01:01 TTDPSSFLGRY $RP_{1637}$, PE-HLA-B*07:02 SPRWYFYYL $NCP_{105}$, Table S3.) and APC conjugated EBV and CMV pentamer/dextramers (APC-HLA-A*02:01 GLCTLVAML BMLF-$1_{259}$, APC-HLA-B*07:02-RPPIFIRRL EBNA-$3A_{247}$, APC-HLA-A*03:01 RLRAEQVK EBNA-$3A_{603}$ or APC-HLA-A*02:01 NLVPMVATV CMV pp65$_{495}$, Table S3.) for 15 min at 37 °C. Surrogate CD3 conjugates for PE and APC were used as reference controls for unmixing either by staining cells (Newcastle) or beads (Sheffield). After pentamer/dextramer staining, samples were washed in PBS and re-suspended in the residual volume (~50 µl). Samples were stained with Live/Dead Zombie NIR (Biolegend, 423106) for 10 min at RT, after which and without washing, the surface antibody cocktail diluted in Brilliant Stain Buffer (Becton Dickinson, 563794) was added for a further 20 min. Single-stain reference controls were stained as per Tables S1 and S2 on $2-5 \times 10^5$ cells/mL or UltraComp eBeads™ (ThermoFisher, 01-2222-42). The Live/Dead Zombie NIR reference control was prepared by staining a 50/50 mixture of live and heat-inactivated (56 °C, 7 min) PBMCs. Following the extracellular staining, samples prepared in Sheffield were also stained for granzyme B using the BD Cytofix/Cytoperm™ Kit (Cat. No. 554714) according to the manufacturer's instructions. Samples and reference controls in Newcastle were washed in PBS after extracellular staining and fixed for 20 min in 2% formaldehyde at RT. After fixation, samples were washed in FACS buffer and re-suspended in an appropriate volume for acquisition on a CyTEK AURORA 3 L (Sheffield) or 5 L (Newcastle) system. Data were unmixed using a combination of bead and cell reference controls, as outlined in Tables S1 and S2, and post-unmixing compensation applied where required.

### Data analysis

Statistical analysis was performed using GraphPad Prism 9.4.1 for Windows. Comparisons between continuous data from SARS-CoV-2-naive and previously-infected groups were performed using Mann–Whitney tests or Kruskal–Wallis tests with Dunn's post hoc test for multiple comparisons, and categorical data compared with the Fisher's exact test. Pair-wise comparisons within groups were performed using Wilcoxon matched-pairs signed-rank tests. Principal component analysis (PCA) was performed using SIMON software version 0.2.1 (https://genular.org)[42]. Before PCA was performed, the data was pre-processed (center/scale), missing values were median imputed, and variables with fewer than 5 unique values were removed. The sex, site, previous infection status, and time points were used as grouping variables and thus were not included in the analysis. The quality of individuals and variable representations (cos2), variable correlations, and contributions (percentage) of the top 10 variables from the first two principal components (PC1 and PC2) were calculated. The cos2 value (square cosine, squared coordinates) is used to measure how well a variable is represented on a graph. A high cos2 means that the variable is well-represented and is positioned near the edge of a circle. A low cos2 means that the variable is not well-represented and is positioned closer to the center of the circle. The sum of cos2 values for a variable on all principal components is equal to one[43,44].

Gating analysis of flow cytometry data was performed in SpectroFlo® (version 3.0.0). Data were gated for live, CD3+CD8+ pentamer/dextramer+ populations from twelve batches across two different sites (Sheffield, Newcastle) (Fig. S7). HLA mismatched pentamer/dextramer stains were used to determine gating for positive events (Fig. S8), and these pentamer/dextramer+ populations were sorted into memory T-cell subsets based on their patterns of expression of the markers CD45RO, CCR7, CD28 and CD95 (Fig. S9). A pentamer/dextramer event rate of ≥20 (2-dimensional gating, combined data from both sites) or ≥0.01% of the total CD8+ population (multi-dimensional clustering, data from sites analyzed separately) was used as a threshold to take forward populations for further analysis.

Multi-dimensional analysis of flow cytometry data was undertaken in R using the packages readxl, CATALYST, cowplot, flowCore, scater, SingleCellExperiment, openxlsx, and ggpubr. FCS data from multimer-positive gates were transformed using a cofactor of 150 and FlowSOM clustering applied to all channel markers present on pentamer/dextramer-specific cells, a maxK of 10, and a random seed. Dimension reduction was performed using UMAP. Statistical comparison of marker expression levels was made using Kruskal–Wallis, with $t$-tests for individual comparisons, and adjusted $p$-values displayed.

### Reporting summary

Further information on research design is available in the Nature Portfolio Reporting Summary linked to this article.

## Data availability

The full dataset used in this study has been made available at the Open Science Framework, https://doi.org/10.17605/OSF.IO/9TSZ6[45]. The FCS data used in the UMAP and marker expression analysis have been deposited in the Zenodo databases https://doi.org/10.5281/zenodo.8045040 and https://doi.org/10.5281/zenodo.8045107[46,47]. Source data are provided with this paper.

## Code availability

The code used to undertake the UMAP and marker expression analysis in Figs. S4–6 has been deposited in the Zenodo databases https://doi.org/10.5281/zenodo.8045040 and https://doi.org/10.5281/zenodo.8045107[46,47].

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

## Acknowledgements

This work was funded in part by the UK Department of Health and Social Care as part of the PITCH (Protective Immunity from T cells to Covid-19 in Health workers) Consortium, by the UKRI as part of "Investigation of proven vaccine breakthrough by SARS-CoV-2 variants in established UK healthcare worker cohorts: SIREN consortium & PITCH Plus Pathway" MR/W02067X/1 and UKRI as part of "PITCH2 - Protective Immunity through T Cells in Healthcare workers 2" MR/X009297/1, with contributions from UKRI/NIHR through the UK Coronavirus Immunology Consortium (UK-CIC), the Huo Family Foundation and The National Institute for Health Research (UKRIDHSC COVID-19 Rapid Response Rolling Call, Grant Reference Number COV19- RECPLAS). E.B. and P.K. are NIHR Senior Investigators and P.K. is funded by NIH (U19 I082360) and this work is funded in part by the Wellcome Trust (WT222426/Z/21/Z). S.J.D. is funded by an NIHR Global Research Professorship (NIHR300791). R.P.P. is funded by a Career Re-entry Fellowship (204721/Z/16/Z). C.J.A.D. was supported by fellowships from Wellcome (211153/Z/18/Z) and the Medical Research Council (MR/X001598/1). This study was supported by the NIHR Newcastle Clinical Research Facility. T.D., J.M., and G.S. are funded by the Chinese Academy of Medical Sciences (CAMS) Innovation Fund for Medical Science (CIFMS), China (grant number: 2018-I2M-2-002), Schmidt Futures, the Red Avenue Foundation, and the Oak Foundation. The Wellcome Centre for Human Genetics is supported by the Wellcome Trust (grant 090532/Z/09/Z). L.T. is supported by the Wellcome Trust (grant number 205228/Z/16/Z) and the National Institute for Health Research Health Protection Research Unit (NIHR HPRU) in Emerging and Zoonotic Infections (NIHR200907) at the University of Liverpool in partnership with UKHSA, in collaboration with Liverpool School of Tropical Medicine and the University of Oxford. D.G.W. is supported by an NIHR Advanced Fellowship in Liverpool. M.C., S.L., L.T., and T.T. are supported by U.S. Food and Drug Administration Medical Countermeasures Initiative contract 75F40120C00085. The Sheffield Teaching Hospitals Observational Study of Patients with Pulmonary Hypertension, Cardiovascular, and Other Respiratory Diseases (STH-ObS) was supported by the British Heart Foundation (PG/11/116/29288). The STH-ObS Chief Investigator Allan Laurie is supported by a British Heart Foundation Senior Basic Science Research Fellowship (FS/18/52/33808). We gratefully acknowledge financial support from the UK Department of Health and Social Care via the Sheffield NIHR Clinical Research Facility award to the Sheffield Teaching Hospitals Foundation NHS Trust. The Wellcome Trust grant is acknowledged as UNS104697 Spectral Cytometry for profiling single cells of the immune system in health and disease to A Filby/NUFCCF. The views expressed are those of the author(s) and not necessarily those of the NHS, the NIHR, the Department of Health, or Public Health England. For the purpose of Open Access, the author has applied a CC BY public copyright licence to any Author Accepted Manuscript version arising from this submission.

## Author contributions

Conceptualization: T.d.S., S.J.D., L.T., P.K., C.J.A.D., R.P.P., S.L.R.J., M.C., A.R., E.B. Methodology: T.d.S., S.J.D., L.T., P.K., C.J.A.D., R.P.P., M.C., A.R., E.B., S.L., H.H., G.S. Formal analysis: H.H., A.R.N., S.L., C.L., A.T., A.A., B.K., M.G.R., T.d.S., R.P.P. Investigation: H.H., A.R.N., S.L., C.L., A.T., A.A., B.K., J.K.T., T.T., P.Z., M.G.R., P.S., M.S., P.A., I.N., M.A., N.A.B., J.M.N., L.G., S.S., I.G., T.R., S.C.M., L.M.H., S.L.D., S.B. Resources: A.B., L.T., E.B. Data curation: H.H., T.d.S., R.P.P. Writing—original draft: H.H., T.d.S.. Writing—review and editing: H.H., A.R.N., S.L., A.A., N.A.B., S.B., T.D., E.B., L.T., S.L.R.J., M.C., C.J.A.D., P.K., S.J.D., R.P.P., T.d.S. Visualization: H.H., A.A., A.R.N., R.P.P., T.d.S. Supervision: C.L., B.K., A.A., T.D., J.M., T.L., D.W., V.H., S.H., E.B., G.S., A.R., L.T., S.L.R.J., M.C., C.J.A.D., P.K., S.J.D., R.P.P., T.d.S. Funding acquisition: P.K., S.J.D., L.T., T.d.S., C.J.A.D., A.R., S.H., V.H.

## Competing interests

S.J.D. is a Scientific Advisor to the Scottish Parliament on COVID-19 for which she receives a fee. Oxford University has entered a joint COVID-19 vaccine development partnership with AstraZeneca. G.S. sits on the GSK Vaccines Scientific Advisory Board and is a founder member of RQ Biotechnology. The remaining authors declare no competing interests.

## Additional information

Hailey Hornsby [1,29], Alexander R. Nicols [2,29], Stephanie Longet [3,4,29], Chang Liu [4,29], Adriana Tomic [5,6,7,8], Adrienn Angyal [1], Barbara Kronsteiner [9,10], Jessica K. Tyerman [2], Tom Tipton [3,4], Peijun Zhang [1], Marta Gallis [1], Piyada Supasa [4], Muneeswaran Selvaraj [4], Priyanka Abraham [9,10], Isabel Neale [9,10], Mohammad Ali [9,10], Natalie A. Barratt [1], Jeremy M. Nell [2,11], Lotta Gustafsson [1,12], Scarlett Strickland [1,12], Irina Grouneva [1], Timothy Rostron [13], Shona C. Moore [14], Luisa M. Hering [14], Susan L. Dobson [14], Sagida Bibi [8], Juthathip Mongkolsapaya [4,15], Teresa Lambe [8,15], Dan Wootton [14,16,17], Victoria Hall [18,19], Susan Hopkins [18,19,20], Tao Dong [13,15,21], Eleanor Barnes [21,22], Gavin Screaton [4,15], The PITCH Consortium*, Alex Richter [23,24,30], Lance Turtle [14,25,30], Sarah L. Rowland-Jones [1,12,30], Miles Carroll [3,4,30],

Christopher J. A. Duncan[2,11,30], Paul Klenerman[9,22,26,30] ✉, Susanna J. Dunachie [9,10,22,27,30], Rebecca P. Payne [2,30] & Thushan I. de Silva [1,12,28,30] ✉

[1]Division of Clinical Medicine, School of Medicine and Population Health, The University of Sheffield, Sheffield, UK. [2]Translational and Clinical Research Institute, Immunity, and Inflammation Theme, Newcastle University, Newcastle, UK. [3]Pandemic Sciences Institute, Nuffield Department of Medicine, University of Oxford, Oxford, UK. [4]Wellcome Centre for Human Genetics, Nuffield Department of Medicine, University of Oxford, Oxford, UK. [5]National Emerging Infectious Diseases Laboratories, Boston University, Boston, MA, USA. [6]Department of Microbiology, Boston University School of Medicine, Boston, MA, USA. [7]Department of Biomedical Engineering, Boston University, Boston, MA, USA. [8]Oxford Vaccine Group, Department of Paediatrics, University of Oxford, Oxford, UK. [9]Peter Medawar Building for Pathogen Research, Nuffield Dept. of Clinical Medicine, University of Oxford, Oxford, UK. [10]NDM Centre For Global Health Research, Nuffield Dept. of Clinical Medicine, University of Oxford, Oxford, UK. [11]Department of Infection and Tropical Medicine, Newcastle upon Tyne Hospitals NHS Foundation Trust, Newcastle upon Tyne, UK. [12]Sheffield Teaching Hospitals NHS Foundation Trust, Sheffield, UK. [13]MRC Human Immunology Unit, MRC Weatherall Institute of Molecular Medicine, University of Oxford, Oxford, UK. [14]NIHR Health Protection Research Unit in Emerging and Zoonotic Infections, Institute of Infection, Veterinary and Ecological Sciences, University of Liverpool, Liverpool, UK. [15]Chinese Academy of Medical Science (CAMS) Oxford Institute (COI), University of Oxford, Oxford, UK. [16]Liverpool University Hospitals NHS Foundation Trust, Liverpool, UK. [17]Institute of Infection, Veterinary and Ecological Sciences, University of Liverpool, Liverpool, UK. [18]UK Health Security Agency, London, UK. [19]Faculty of Medicine, Department of Infectious Disease, Imperial College London, London, UK. [20]NIHR Health Protection Research Unit in Healthcare Associated Infection and Antimicrobial Resistance, University of Oxford, Oxford, UK. [21]Nuffield Department of Medicine, University of Oxford, Oxford, UK. [22]Oxford NIHR Biomedical Research Centre and Oxford University NHS Foundation Trust, Oxford, UK. [23]Institute for Immunology and Immunotherapy, College of Medical and Dental Science, University of Birmingham, Birmingham, UK. [24]University Hospitals Birmingham NHS Foundation Trust, Birmingham, UK. [25]Tropical & Infectious Disease Unit, Liverpool University Hospitals NHS Foundation Trust (member of Liverpool Health Partners), Liverpool, UK. [26]Translational Gastroenterology Unit, University of Oxford, Oxford, UK. [27]Mahidol-Oxford Tropical Medicine Research Unit, Mahidol University, Bangkok, Thailand. [28]Vaccines and Immunity Theme, Medical Research Council Unit The Gambia at the London School of Hygiene and Tropical Medicine, Banjul, The Gambia. [29]These authors contributed equally: Hailey Hornsby, Alexander R. Nicols, Stephanie Longet, Chang Liu. [30]These authors jointly supervised this work: Alex Richter, Lance Turtle, Sarah L. Rowland-Jones, Miles Carroll, Christopher J. A. Duncan, Paul Klenerman, Susanna J. Dunachie, Rebecca P. Payne, Thushan I. de Silva. ✉e-mail: paul.klenerman@ndm.ox.ac.uk; t.desilva@sheffield.ac.uk

## The PITCH Consortium

Hailey Hornsby [1,29], Alexander R. Nicols [2,29], Stephanie Longet [3,4,29], Chang Liu[4,29], Adrienn Angyal[1], Barbara Kronsteiner [9,10], Jessica K. Tyerman[2], Tom Tipton[3,4], Peijun Zhang[1], Piyada Supasa[4], Priyanka Abraham[9,10], Isabel Neale [9,10], Mohammad Ali[9,10], Natalie A. Barratt [1], Jeremy M. Nell[2,11], Lotta Gustafsson[1,12], Scarlett Strickland[1,12], Irina Grouneva[1], Shona C. Moore [14], Luisa M. Hering[14], Susan L. Dobson[14], Sagida Bibi[8], Juthathip Mongkolsapaya[4,15], Teresa Lambe [8,15], Dan Wootton[14,16,17], Victoria Hall [18,19], Susan Hopkins[18,19,20], Eleanor Barnes [21,22], Gavin Screaton [4,15], Alex Richter[23,24,30], Lance Turtle [14,25,30], Sarah L. Rowland-Jones[1,12,30], Miles Carroll[3,4,30], Christopher J. A. Duncan[2,11,30], Paul Klenerman [9,22,26,30] ✉, Susanna J. Dunachie [9,10,22,27,30], Rebecca P. Payne [2,30] & Thushan I. de Silva [2,30]

A full list of members and their affiliations appears in the Supplementary Information.

