## [Peer Review File · Nature Communications]

Omicron infection following vaccination enhances a broad spectrum of immune responses dependent on infection historyREVIEWER COMMENTS

Reviewer #1 (Remarks to the Author):

This is a straightforward and clearly written manuscript of a well-designed study that provides insights into the hybrid immunity induced by omicron breakthrough infections in triple-vaccinated healthcare workers with and without previous SARS-CoV-2 infection. In contrast to other studies, this one includes mucosal antibodies and T-cell responses. Findings show that although the NAb responses are lower in previously infected vs naïve individuals, other immune responses are higher in previously infected, and overall immunity after omicron breakthrough seems to be robust.

I do not have major concerns but the manuscript could be improved in some aspects:

- Some abbreviations need to be defined the first time they appear in the manuscript such as PITCH and MSD.
- The information on the interval times between 2nd and 3rd vaccine doses is not reported.
- Why the median time between 1st and 2nd was 9.6 weeks apart if the recommended schedule for BNT162b2 was 3 weeks?
- In table 1, "(sequence confirmed)" should be removed since sequences were confirmed only for some (as indicated in the footnote).
- The first time nAb are mentioned in the results section, it would be informative to mention the type of assay used.
- When reporting T cell responses from Fig 1, I think it is important to be more specific and say that are "IFN-g" T cell responses. It should be added to the plots' titles too.
- It would be helpful for the readers to include in all figure legends "after vaccination" or "vaccinated". For example Figure 1. Comparison of immune responses after vaccination prior to omicron infection in SARS-CoV-2-naïve and previously-infected individuals.
- Figure 2. Impact of omicron infection on plasma neutralizing and binding antibodies in vaccinated SARS-CoV-2-naïve and previously-infected individuals.
- Why different methods were used to measure Spike and nucleocapsid-specific IgG?
- In Fig 1 legend, only Mann-Whitney U test is said that had been used. For paired comparisons, the Wilcoxon signed-rank test should be used. It looks like an oversight of the authors for this legend as in others both tests are indicated.
- In relation to the sentence: Significant heterogeneity was evident in individual plasma NAb trajectories to BA.1 and BA.2 (Figure 2B & 2C), with antibody responders and non-responders seen in both groups. BA.5 neutralizing antibodies are not mentioned nor shown. Were they not evaluated? Some explanation is needed if they were not measured or if measured, then what the results were.
- What does "matrix" mean in the following sentence? "In contrast to spike responses, T cell responses to a matrix and nucleocapsid peptide pool increased in both SARS-CoV-2 naïve (11.8-fold, $p < 0.0001$) and previously-infected (2.0-fold, $p = 0.0004$) individuals"
- Regarding the significant increases in ELISpot responses to peptide pools representing ancestral non-structural proteins observed in SARS-CoV-2 naïve individuals after omicron infection, not were

only overall low, but also most were below the positivity threshold.

- Overall, the font size in figures is too small, and very difficult to read if the manuscript is printed.
- In Fig 5C, the color scale is confusing. Why there are two colors?
- I think an explanation of the meaning of “quality of the representation (\cos^2)” is needed for readers not familiar with PCA, and particularly because PC loadings are more often shown than \cos^2 .
- The full gating strategy should be shown in supplementary materials.
- When discussing the hypotheses for the lower nAb in previously infected individuals, another possibility would be that IgG subclasses induced in previously infected are those with other antibody effector functions than neutralization.
- An additional limitation that is not stated in the discussion, and that affects generalisability of findings, is that is a cohort of healthcare workers, so relatively young and healthy individuals that do not represent the whole population.
- The conclusion (last sentence of the discussion) that mucosal and non-spike responses should also be prioritized as targets to broaden the responses generated by the next generation of vaccines is not supported by the findings of the study. I agree that mucosal responses should be prioritized to induce protection against infection, but I think is unclear the benefit of targeting non-spike antigens.
- A description of how antibody positivity thresholds are calculated should be included in the manuscript.
- Finally, information on the origin of the peptide pools should be added. Pools of CMV, EBV and influenza peptides (CEF) and phytohemagglutinin (PHA) were also used at 2 $\mu\text{g}/\text{mL}$?

Reviewer #2 (Remarks to the Author):

The authors present a thorough analysis of a cohort of omicron breakthrough infections and thereby contribute a considerable data set to the field of SARS-CoV-2 long term immunity. I recommend the following revisions to the manuscript:

Ad Table 1: Please reformat the entire table to make it more readable, e.g. some highlighting with thicker lines, abbreviating some of the titles and correcting the vertical positions of the numbers would improve it a lot. I would suggest using a different software which allows for nicer formatting (LaTeX or similar), but if this is not available putting some more effort into the table should help a lot to make it more presentable.

Ad Fig. 2B/2C: Why were the individual plasma Nab trajectories for BA.5 omitted here? If they were deemed uninteresting (even though heterogeneity looks similar to BA.1 and BA.2 in Fig.2A), please state and justify that decision in the text. Also, since it's necessary to graph this type of data in \log_{10} please mention the range of fold change for each group in the text for the reader to get a better idea of the differences described.

Ad Fig. 2D: Please discuss implications of the fact that in the naïve post-omicron group 22.6% of participants did not mount an N response after omicron infection. Can you compare that to data from the literature, especially on omicron-specific N responses of unvaccinated individuals?

Ad Fig. 3C: It is not clear from the description why the authors switched to a surrogate neutralization assay when the live virus neutralization assay was available, especially considering the wide spread of the results in the surrogate assay. Please explain the reasoning here: Does the collection method yield too little of each sample or are other components in the nasal lining fluid toxic for the Vero cells?

Against both of these reasons it could be argued that predilutions are usually necessary for samples from triple vaccinated individuals. It would be great to see if a small batch of samples analyzed via the live virus neutralization assay yield similar fold differences.

Ad Fig. 4A/B: Why were the experiments not performed with BA.5 peptide pools (which are commercially available)?

Ad Fig. 6: All subfigures are very low resolution, so that the small size can't be remedied by zooming in, since the graphs become blurry upon magnification. Please revise before resubmission

Ad Fig. 6A: Even being aware that epitope-specific occur at extremely low frequencies and even though it's hard to identify some of the mean bars, the differences are very small between groups. This should at least be mentioned in the text as a caveat to any result interpretation (lines 362-364).

Ad lines 495-497: Please expand on the statement that your "cohort [is] enriched for previously-infected healthcare workers with low mucosal immunity". Why do you expect this bias in the cohort? On that note, please specify what kind of health care workers, are they at increased risk of infection, especially with the predominantly female participants please specify whether in the data analysis throughout the paper there is any significant differences between participant groups with and without patient contact, work in intensive care/COVID units. If there are differences can you speculate whether exposure dosage has anything to do with the observed strength in immune responses or whether repeated exposure could influence the outcome in immunity? Can you exclude repeated infection with ancestral variants in your cohort?

Reviewer #3 (Remarks to the Author):

Recently, several studies observed a weak neutralizing antibody response to SARS-CoV-2 omicron VOCs in individuals that had an ancestral/alpha/beta SARS-CoV-2 infection prior to (triple) SARS-CoV-2 vaccination (Blom et al., *Lancet Infect. Dis* 2022; Reynolds et al., *Science* 2022). These findings questioned efficient immunity against infection with upcoming SARS-CoV-2 omicron VOCs. Hornsby, Nicols, Longet et al. now addressed this issue in a more comprehensive manner. They used a quite large and very-well characterized cohort of 56 individuals that received triple mRNA vaccination but were SARS-CoV-2-naïve as well as 38 individuals with an early (pre-omicron) infection prior to triple vaccination. In these two groups, the authors studied neutralizing antibodies, plasma as well nasal spike- and nucleocapsid-specific antibodies, and spike- as well as nucleocapsid-/membrane-specific T cells. In these analyses, they indeed confirmed an inferior neutralizing antibody response of individuals with previous SARS-CoV-2 infection compared to naïve patients, however, nasal antibody responses were comparable in the two groups, and T cell responses were boosted and remained superior in the group with previous infection. In sum, these results indicate that patients with a history of pre-omicron infection and triple mRNA vaccination still respond to omicron infection and display a boosted nasal antibody response as well as T cell response and are thus likely well prepared to respond to infection with future omicron VOCs.

Overall, this is an important and well performed study. There are, however, some limitations that need attention:

1. Title: "Diverse repertoire" suggests that the immune response repertoire has been studied. This is not the case. Just looking at spike- versus nucleocapsid-/membrane specific antibody and T cell responses is not an analysis of the immune response repertoire. The authors acknowledge this limitation ("It is likely that individual epitope-specific differences may exist between groups, including those affected by mutations in omicron lineage viruses", lines 560/561). The title needs to be phrased more clearly.
2. Post-omicron sample were taken a median 30 days after infection. The booster effect of repetitive vaccination (e.g., third vaccine dose) has been described to last for 30-60 days before spike-specific T cells decrease to baseline levels. Please show (and if relevant discuss) the effect of the interval between vaccination and infection more clearly.
3. Figure 3C: For the previously-infected group, ACE2 inhibition does not change (1.0x) between pre-

and post-omicron analysis, however, the p value reaches significance (0.04). This discrepancy may be due to using the median for calculating the factor, but it is confusing and should be resolved/explained.

4. The phenotypical T cell analyses displayed in Fig. 6C and D are quite rude (memory phenotypes and granzyme expression only), while some more advanced analyses are only shown in the supplementary figures and are not really mentioned/discussed in the results and discussion section. In addition, all these figures (Fig. 6C/D, Fig. S4-6) are just too tiny to recognize anything. Data on T cell phenotypes should be displayed adequately, or should be deleted from the manuscript. In the current version, the term "detailed characterisation" (line 520) is a vast exaggeration of the described analyses.

REVIEWER COMMENTS

We would very much like to thank the reviewers for their time and for their helpful and insightful comments on our manuscript. We have provided our point by point responses below.

Reviewer #1 (Remarks to the Author):

This is a straightforward and clearly written manuscript of a well-designed study that provides insights into the hybrid immunity induced by omicron breakthrough infections in triple-vaccinated healthcare workers with and without previous SARS-CoV-2 infection. In contrast to other studies, this one includes mucosal antibodies and T-cell responses. Findings show that although the NAb responses are lower in previously infected vs naïve individuals, other immune responses are higher in previously infected, and overall immunity after omicron breakthrough seems to be robust.

I do not have major concerns but the manuscript could be improved in some aspects:• Some abbreviations need to be defined the first time they appear in the manuscript such as PITCH and MSD.

Thank you for pointing this out, we have made sure to introduce these abbreviations the first time they appear.

- The information on the interval times between 2nd and 3rd vaccine doses is not reported.

We have now included this information in the text as follows:

“Participants received their 2nd and 3rd vaccine doses a median of 34.9 weeks apart (IQR 32.4-37.4).”

- Why the median time between 1st and 2nd was 9.6 weeks apart if the recommended schedule for BNT162b2 was 3 weeks?

The schedule for the first 2 doses of BNT162b2 was changed in the UK on December 31 2020, from 3 weeks to up to 12 weeks to ensure that the maximum number of people could receive their first dose quickly. The majority of our cohort were not vaccinated at this point and so received their doses on the extended schedule. To make this clear we have added a sentence to the text explaining it, and have referenced the UKHSA statement announcing the change in schedule:

“All individuals had received their 1st and 2nd vaccine doses a median of 9.6 weeks apart (IQR 8.9-10.9) in line with UK Health Security Agency (UKHSA) guidelines which increased the recommended dosing interval from 3 weeks to up to 12 weeks in December 2021 ¹⁴.”

Ref 14 - Department of Health and Social Care UK. Optimising the COVID-19 vaccination programme for maximum short-term impact.
<https://www.gov.uk/government/publications/prioritising-the-first-covid-19-vaccine-dose-icvi-statement/optimising-the-covid-19-vaccination-programme-for-maximum-short-term-impact> (2021).

- In table 1, “(sequence confirmed)” should be removed since sequences were confirmed only for some (as indicated in the footnote).

The numbers given in brackets were intended to show the numbers for which sequence was confirmed, however this was not very clear and we have now reworked the table to improve readability.

- The first time nAb are mentioned in the results section, it would be informative to mention the type of assay used.

We have added a reference to the FRNT assay upon first mention of NAb in results.

- When reporting T cell responses from Fig 1, I think it is important to be more specific and say that they are “IFN- γ ” T cell responses. It should be added to the plots’ titles too.

We have added IFN- γ to the titles of all T cell plots.

- It would be helpful for the readers to include in all figure legends “after vaccination” or “vaccinated”. For example Figure 1. Comparison of immune responses after vaccination prior to omicron infection in SARS-CoV-2-naive and previously-infected individuals. - Figure 2. Impact of omicron infection on plasma neutralizing and binding antibodies in vaccinated SARS-CoV-2-naive and previously-infected individuals.

Thank you for pointing this out, we have made this change to all figure legends to make it immediately clear that these are data from a vaccinated cohort.

- Why different methods were used to measure Spike and nucleocapsid-specific IgG?

The MSD plates we used to assay the anti-spike antibodies did not include an anti-N spot, and due to the high cost of MSD kits and the fact that we already had an in-house anti-N ELISA set up, validated and described in our previous publications we decided to use this to assay the anti-N antibody levels in plasma.

- In Fig 1 legend, only Mann-Whitney U test is said that had been used. For paired comparisons, the Wilcoxon signed-rank test should be used. It looks like an oversight of the authors for this legend as in others both tests are indicated.

All Figure 1. plots compare the response of the SARS-CoV-2 naïve and previously infected participants in various assays at the *pre-infection* timepoint only, so no paired comparisons were made – thus only the Mann-Whitney test was used and not the Wilcoxon. We feel this is clearly explained in the figure legend.

- In relation to the sentence: Significant heterogeneity was evident in individual plasma NAb trajectories to BA.1 and BA.2 (Figure 2B & 2C), with antibody responders and non-responders seen in both groups. BA.5 neutralizing antibodies are not mentioned nor shown. Were they not evaluated? Some explanation is needed if they were not measured or if measured, then what the results were.

We have now added the BA.5 NAb trajectories and condensed this into 1 Figure (2B) so all these data can be seen.

- What does “matrix” mean in the following sentence? “In contrast to spike responses, T cell responses to a matrix and nucleocapsid peptide pool increased in both SARS-CoV-2 naive (11.8-fold, $p < 0.0001$) and previously-infected (2.0-fold, $p = 0.0004$) individuals”.

Thank you for spotting this mistake – matrix has been changed to membrane in this sentence.

- Regarding the significant increases in ELISpot responses to peptide pools representing ancestral non-structural proteins observed in SARS-CoV-2 naive individuals after omicron infection, not were only overall low, but also most were below the positivity threshold.

We have added a comment in the text to draw attention to this:

“Significant increases in ELISpot responses to peptide pools representing ancestral non-structural proteins (...) were seen in SARS-CoV-2 naive individuals after omicron infection (Figure S2), although overall responses in each pool were low and often remained below the positivity threshold.”

- Overall, the font size in figures is too small, and very difficult to read if the manuscript is printed.

Thank you for this input, we have reformatted all plots to increase font size as well as size of individual plots and improved readability in various ways.

- In Fig 5C, the color scale is confusing. Why there are two colors?

and

- I think an explanation of the meaning of “quality of the representation (\cos^2)” is needed for readers not familiar with PCA, and particularly because PC loadings are more often shown than \cos^2 .

In Figure 5C, the colours exist on a scale from blue to red which represent low to high \cos^2 contribution of each of the immune parameters, the colour legend is included above the plot. In order to further clarify the definition of \cos^2 we have added the following text to the methods section, and added two references:

“The \cos^2 value (square cosine, squared coordinates) is used to measure how well a variable is represented on a graph. A high \cos^2 means that the variable is well-represented and is positioned near the edge of a circle. A low \cos^2 means that the variable is not well-represented and is positioned closer to the center of the circle. The sum of \cos^2 values for a variable on all principal components is equal to one^{40, 41}.”

Ref 41 - Abdi, Hervé, and Lynne J. Williams. 2010. “Principal Component Analysis.” John Wiley and Sons, Inc. WIREs Comp Stat 2: 433–59.

<http://staff.ustc.edu.cn/~zwp/teach/MVA/abdi-awPCA2010.pdf>.

Ref 42 - Husson, Francois, Sebastien Le, and Jérôme Pagès. 2017. Exploratory Multivariate Analysis by Example Using R. 2nd ed. Boca Raton, Florida: Chapman; Hall/CRC.

<http://factominer.free.fr/bookV2/index.html>.

- The full gating strategy should be shown in supplementary materials.

We have added plots to the supplementary showing the gating strategies used for this study (Figures S7-S9).

- When discussing the hypotheses for the lower nAb in previously infected individuals, another possibility would be that IgG subclasses induced in previously infected are those with other antibody effector functions than neutralization.

Thank you for suggesting this hypothesis. However, the total binding IgG antibody titres after omicron in the previously infected group do not go up (in fact the median titres reduce compared to the dose 3 + 28 day titre, which does not of course take account of waning prior to omicron infection). We therefore do not observe more binding antibody induction than NAb induction which would be the basis of this hypothesis. We have therefore not mentioned this in the discussion.

- An additional limitation that is not stated in the discussion, and that affects generalisability of findings, is that is a cohort of healthcare workers, so relatively young and healthy individuals that do not represent the whole population.

We have added a line to our discussion to call attention to this limitation:

“Additionally, our cohort of healthcare workers are relatively young and healthy, and so not entirely representative of the general population.”

- The conclusion (last sentence of the discussion) that mucosal and non-spike responses should also be prioritized as targets to broaden the responses generated by the next generation of vaccines is not supported by the findings of the study. I agree that mucosal responses should be prioritized to induce protection against infection, but I think is unclear the benefit of targeting non-spike antigens.

Recent data demonstrate that non-spike T cell epitope induction is specifically being targeted in a new mRNA vaccine strategy, with animal data showing protection from severe disease. Human trials are being planned to combine this vaccine with the BA4/5 spike mRNA construct in use. We have therefore left this statement in and referenced this recent manuscript.

Ref 38 - Arieta, C. M. *et al.* The T-cell-directed vaccine BNT162b4 encoding conserved non-spike antigens protects animals from severe SARS-CoV-2 infection. *Cell* S0092867423004038 (2023) doi:10.1016/j.cell.2023.04.007.

- A description of how antibody positivity thresholds are calculated should be included in the manuscript.

The text has been amended to give a brief description of how thresholds were determined, and to reference our previous work which describes this process in greater detail:

“As previously reported³⁹, serostatus of samples was determined based on a threshold selected to maximise sensitivity, validated using sera from PCR-confirmed SARS-CoV-2 convalescent patients and pre-2019 samples. Samples considered negative according to this threshold were assigned a value of 1.04 BAU/mL, or half the value of the lowest point on the standard curve.”

Ref 39 - Colton, H. et al. Risk factors for SARS-CoV-2 seroprevalence following the first pandemic wave in UK healthcare workers in a large NHS Foundation Trust. Wellcome Open Res. 6, 220 (2022).

• Finally, information on the origin of the peptide pools should be added. Pools of CMV, EBV and influenza peptides (CEF) and phytohemagglutinin (PHA) were also used at 2 ug/mL?

We have added the concentrations used for the CEF/PHA controls to the methods section. Pooled peptides were obtained from Mimotopes (<http://www.mimotopes.com>). We have also added this information to the text.

Reviewer #2 (Remarks to the Author):

The authors present a thorough analysis of a cohort of omicron breakthrough infections and thereby contribute a considerable data set to the field of SARS-CoV-2 long term immunity. I recommend the following revisions to the manuscript:

Ad Table 1: Please reformat the entire table to make it more readable, e.g. some highlighting with thicker lines, abbreviating some of the titles and correcting the vertical positions of the numbers would improve it a lot. I would suggest using a different software which allows for nicer formatting (LaTeX or similar), but if this is not available putting some more effort into the table should help a lot to make it more presentable.

Thank you for your recommendations here, we have reformatted the table to improve readability, we hope you find this is an improved way of displaying our cohort details.

Ad Fig. 2B/2C: Why were the individual plasma Nab trajectories for BA.5 omitted here? If they were deemed uninteresting (even though heterogeneity looks similar to BA.1 and BA.2 in Fig.2A), please state and justify that decision in the text. Also, since it's necessary to graph this type of data in log10 please mention the range of fold change for each group in the text for the reader to get a better idea of the differences described.

We have now added the BA.5 NAb trajectories and condensed this into 1 Figure (2B) so all this data can be seen. We have also added data on the range of fold change against ancestral and omicron viruses in our groups:

“Fold change from pre- to post-omicron infection in naive individuals ranged from 0.44x to 29.3x against ancestral SARS-CoV-2, and from 0.51x to 156.4x against omicron subvariants, while in the previously infected fold change ranged from 0.50x to 4.9x against ancestral virus, and from 0.19x to 16.4x against omicron subvariants.”

Ad Fig. 2D: Please discuss implications of the fact that in the naïve post-omicron group 22.6% of participants did not mount an N response after omicron infection. Can you compare that to data from the literature, especially on omicron-specific N responses of unvaccinated individuals?

Our data are consistent with other UKHSA cohort data showing that N IgG induction is low in breakthrough infections in vaccinees, even after 2 doses of vaccine and delta breakthrough infections. We have added a brief discussion point and referenced this manuscript. This is significantly lower than N seroconversion in unvaccinated individuals.

“While 22.6% of naive participants did not mount a detectable N-specific IgG response upon omicron infection, this is consistent with UKHSA data showing that breakthrough infections in fully vaccinated individuals result in poor induction of N-specific IgG, possibly due to the lower disease severity experienced ³².”

Ref 32 - Whitaker, H. J. et al. Nucleocapsid antibody positivity as a marker of past SARS-CoV-2 infection in population serosurveillance studies: impact of variant, vaccination, and choice of assay cut-off. Preprint at <https://doi.org/10.1101/2021.10.25.21264964> (2021).

Ad Fig. 3C: It is not clear from the description why the authors switched to a surrogate neutralization assay when the live virus neutralization assay was available, especially considering the wide spread of the results in the surrogate assay. Please explain the reasoning here: Does the collection method yield too little of each sample or are other components in the nasal lining fluid toxic for the Vero cells? Against both of these reasons it could be argued that predilutions are usually necessary for samples from triple vaccinated individuals. It would be great to see if a small batch of samples analyzed via the live virus neutralization assay yield similar fold differences.

Pilot unpublished data in our laboratory has shown that the signal from mucosal samples in live virus neut assays is too low to detect reliably. We have therefore used the surrogate neutralisation assay (on undiluted samples) from mucosal samples as an alternative (Longet et al Frontiers in Immunology). The sample collection method may also affect how much material is collected, but work done previously by our group has shown that SAM strips (as used in this study) give the greatest yield for antibody measurement (de Silva Journal of Immunological Methods).

Ad Fig. 4A/B: Why were the experiments not performed with BA.5 peptide pools (which are commercially available)?

We have not used commercially available peptides as in our consortium we have consistently used in-house designed sets throughout the pandemic and want to maintain comparability with our prior data (e.g. Angyal et al, Payne et al, Moore et al). Therefore VOC peptide sets are also designed as updates to this original set of backbone peptides. We had BA.1 and BA.2 peptides available at the time of experiments. Furthermore, cell numbers available meant that we had to make some decisions about which peptide sets to prioritise and given that infections were due to BA.1 or BA.2, we felt that responses to these were of greater value than to BA.5 in this manuscript.

Ad Fig. 6: All subfigures are very low resolution, so that the small size can't be remedied by zooming in, since the graphs become blurry upon magnification. Please revise before resubmission

We have split Figure 6 into two figures (now Figures 6 & 7) and increased the sizes of the individual plots to allow better resolution of the details.

Ad Fig. 6A: Even being aware that epitope-specific occur at extremely low frequencies and even though it's hard to identify some of the mean bars, the differences are very small between groups. This should at least be mentioned in the text as a caveat to any result interpretation (lines 362-364).

We have added a mention of the small differences as a caveat to our discussion of these results.

Ad lines 495-497: Please expand on the statement that your “cohort [is] enriched for previously-infected healthcare workers with low mucosal immunity”. Why do you expect this bias in the cohort?

We have reworded the text to make our meaning here clearer:

“...and is perhaps explained by our selection of a cohort of healthcare workers who experienced an omicron infection, thereby potentially enriching for previously-infected individuals at increased risk of breakthrough infection due to low mucosal immunity.”

On that note, please specify what kind of health care workers, are they at increased risk of infection, especially with the predominantly female participants please specify whether in the data analysis throughout the paper there is any significant differences between participant groups with and without patient contact, work in intensive care/COVID units. If there are differences can you speculate whether exposure dosage has anything to do with the observed strength in immune responses or whether repeated exposure could influence the outcome in immunity?

Unfortunately in our healthcare worker cohort we have not collected this level of metadata so do not have place of work, exposure duration to include in any analysis.

Can you exclude repeated infection with ancestral variants in your cohort?

Thank you to the reviewer for bringing up this interesting point. It is certainly possible that a very small number of our convalescent cohort may have had a 2nd ancestral infection prior to their omicron infections. Prior to the advent of delta and omicron variants, this occurrence was very rare in UK healthcare workers (Hall et al.). While we did not have routine and systematic asymptomatic PCR screening in the cohort, healthcare workers in the UK were encouraged to do regular LF and/or PCR tests every week during several stages of the pandemic. We are therefore confident that we would have picked up any repeated infections in our cohort.

Reviewer #3 (Remarks to the Author):

Recently, several studies observed a weak neutralizing antibody response to SARS-CoV-2 omicron VOCs in individuals that had an ancestral/alpha/beta SARS-CoV-2 infection prior to (triple) SARS-CoV-2 vaccination (Blom et al., *Lancet Infect. Dis* 2022; Reynolds et al., *Science* 2022). These findings questioned efficient immunity against infection with upcoming SARS-CoV-2 omicron VOCs. Hornsby, Nicols, Longet et al. now addressed this issue in a more comprehensive manner. They used a quite large and very-well characterized cohort of 56 individuals that received triple mRNA vaccination but were SARS-CoV-2-naïve as well as 38 individuals with an early (pre-omicron) infection prior to triple vaccination. In these two groups, the authors studied neutralizing antibodies, plasma as well nasal spike- and nucleocapsid-specific antibodies, and spike- as well as nucleocapsid-/membrane-specific T cells. In these analyses, they indeed confirmed an inferior neutralizing antibody response of individuals with previous SARS-CoV-2 infection compared to naïve patients, however, nasal antibody responses were comparable in the two groups, and T cell responses were boosted and remained superior in the group with previous infection. In sum, these results indicate that patients with a history of pre-omicron infection and triple mRNA vaccination still respond to omicron infection and display a boosted nasal antibody response as well as T cell response and are thus likely well prepared to respond to infection with future omicron VOCs.

Overall, this is an important and well performed study. There are, however, some limitations that need attention:

1. Title: "Diverse repertoire" suggests that the immune response repertoire has been studied. This is not the case. Just looking at spike- versus nucleocapsid-/membrane specific antibody and T cell responses is not an analysis of the immune response repertoire. The authors acknowledge this limitation ("It is likely that individual epitope-specific differences may exist between groups, including those affected by mutations in omicron lineage viruses", lines 560/561). The title needs to be phrased more clearly.

We have removed the phrase 'diverse repertoire' from our title so that it does not mislead.

2. Post-omicron sample were taken a median 30 days after infection. The booster effect of repetitive vaccination (e.g., third vaccine dose) has been described to last for 30-60 days before spike-specific T cells decrease to baseline levels. Please show (and if relevant discuss) the effect of the interval between vaccination and infection more clearly.

The SIMON analysis (shown in figure 6E) demonstrates the contribution of several variables to the heterogeneity in the data observed, including the interval between booster vaccine and infection, which contributes very little. We have amended the results to make this clearer:

"Plasma antibody and blood T cell responses were the most important factors in immunophenotypic variability, while variables such as age or time between 3rd vaccine dose and omicron infection contributed very little to the variation in our data (Figure 5E)."

3. Figure 3C: For the previously-infected group, ACE2 inhibition does not change (1.0x) between pre- and post-omicron analysis, however, the p value reaches significance (0.04). This discrepancy may be due to using the median for calculating the factor, but it is confusing and should be resolved/explained.

You are correct that this is not an error and is due to the use of median, we have added a note to the text to clarify:

"Note that although the fold change from pre- to post- omicron against BA.2 in the previously infected group is 1.0x, the p-value=0.0425; this is due to the use of median fold change to calculate the statistic."

4. The phenotypical T cell analyses displayed in Fig. 6C and D are quite rude (memory phenotypes and granzyme expression only), while some more advanced analyses are only shown in the supplementary figures and are not really mentioned/discussed in the results and discussion section.

We have added some of the data from the UMAPs plots (Figures 4-6) to this section, highlighting which markers are differentially expressed between our groups.

In addition, all these figures (Fig. 6C/D, Fig. S4-6) are just too tiny to recognize anything. Data on T cell phenotypes should be displayed adequately, or should be deleted from the manuscript.

We have split Figure 6 into two figures (now Figs. 6 & 7) to allow individual plot sizes to be increased for better resolution. We have also increased font sizes on these figures. Figures S4-6 have been reworked to allow the data to be seen properly.

In the current version, the term “detailed characterisation” (line 520) is a vast exaggeration of the described analyses.

Thank you for this feedback, we have removed ‘detailed’ from this sentence so it does not mislead.

REVIEWERS' COMMENTS

Reviewer #1 (Remarks to the Author):

Most of my concerns have been addressed but the font size of most of the figures is still too small and difficult to read. I have no further comments.

Reviewer #2 (Remarks to the Author):

You have taken into consideration all of my earlier comments and have implemented changes wherever possible. I'm satisfied with all of the explanations wherever a change or addition was not possible. I recommend the manuscript in its current form for publication.

Reviewer #3 (Remarks to the Author):

The authors have adequately addressed all my previous comments. Well done!